# Multidimensional Uncertainty Quantification via Optimal Transport

## Abstract

Most uncertainty quantification (UQ) approaches provide a single scalar value as a measure of model reliability. However, different uncertainty measures could provide complementary information on the prediction confidence. Even measures targeting the same type of uncertainty (e.g., ensemble-based and density-based measures of epistemic uncertainty) may capture different failure modes. We take a multidimensional view on UQ by stacking complementary UQ measures into a vector. Such vectors are assigned with Monge-Kantorovich ranks produced by an optimal-transport-based ordering method. The prediction is then deemed more uncertain than the other if it has a higher rank. The resulting *VecUQ-OT* algorithm uses entropy-regularized optimal transport. The transport map is learned on vectors of scores from in-distribution data and, by design, applies to unseen inputs, including out-of-distribution cases, without retraining. Our framework supports flexible non-additive uncertainty fusion (including aleatoric and epistemic components). It yields a robust ordering for downstream tasks such as selective prediction, misclassification detection, out-of-distribution detection, and selective generation. Across synthetic, image, and text data, *VecUQ-OT* shows high efficiency even when individual measures fail.

## 1 Introduction

Uncertainty quantification (UQ) in machine learning is a rapidly growing field (Hüllermeier & Waegeman, 2021), driven by the increasing deployment of artificial intelligence systems in critical applications (Begoli et al., 2019; Kendall & Gal, 2017).

In these applications, it is essential to distinguish between two types of predictive uncertainty. The first is *aleatoric uncertainty*, which arises from inherent randomness in the relationship between covariates $x$ and labels $y$. The second is *epistemic uncertainty*, which reflects limited knowledge of the true data-generating distribution $p(y \mid x)$ and is, in practice, harder to characterize.

Because each source of uncertainty plays a distinct role in downstream tasks, considerable effort has focused on designing estimators that capture these components accurately (Kotelevskii et al., 2025b; Hofman et al., 2024; Schweighofer et al., 2023; 2025; Wimmer et al., 2023). However, the abundance of uncertainty measures creates a practical challenge: practitioners must select a suitable measure for each task at hand (Schweighofer

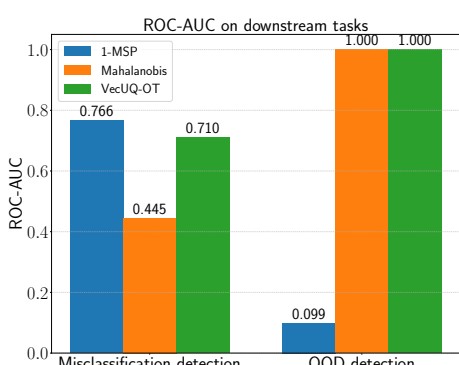

Figure 1: **Complementarity of uncertainty measures.** ROC-AUC on two downstream tasks for two standard scalars and our vector-based method `VecUQ-OT`. Each scalar excels on only one task, whereas `VecUQ-OT` remains robust across both.

et al., 2025). This complicates deployment in typical machine-learning workflows, since even closely related measures (e.g., proxies for a particular source of uncertainty) can behave quite differently.

Ideally, one would leverage multiple measures *simultaneously* to obtain a more informative and robust uncertainty representation across tasks. As a motivating example, consider two downstream problems: misclassification detection and out-of-distribution (OOD) detection (Figure 1; Section A). We focus on two uncertainty measures: $1-$maximum softmax probability (1-MSP) (Hendrycks & Gimpel, 2017) and the Mahalanobis score (Lee et al., 2018). As each individual measure performs well in *only one task* (see ROC-AUC values in Figure 1), the choice of a measure must be problem-specific. In general, even within a single task (e.g., OOD detection), the performance of a particular single UQ measure can vary significantly across datasets.

To overcome this problem, we propose a vector-valued uncertainty representation with multiple components. In our example, these would be two: 1-MSP and the Mahalanobis score. We combine these two one-dimensional uncertainty measures via optimal transport (OT), and denote the resulting composite measure as `VecUQ-OT` (Vector Uncertainty via OT). As seen in Figure 1, `VecUQ-OT` performs robustly for *both tasks*, even when one component is weak. A natural way to pursue such robustness is to stack several one-dimensional measures into a vector and "order" them by the amount of uncertainty they contain. This idea is straightforward, but unlike scalars, vectors lack a canonical order. Recent work on optimal transport and multivariate ranks (Chernozhukov et al., 2017) provides a remedy by inducing a principled ordering over multivariate observations. This paper adapts these ideas to construct and compare *multidimensional uncertainty vectors*.

The main **contributions** of this work can be summarized as follows.

1. We propose representing predictive uncertainty as a *vector* by stacking multiple, complementary uncertainty measures rather than committing to a single scalar. To the best of our knowledge, this is the first UQ framework to treat uncertainty explicitly as a vector.
2. We instantiate a principled way to compare such vectors using ideas from *Monge-Kantorovich (MK) ranks* to induce an order over vectors.
3. We provide a practical implementation that uses *entropy-regularized optimal transport* (Cuturi, 2013) with MK ranks, calibrated on in-distribution scores. The procedure is designed to generalize beyond the calibration set and, empirically, performs well even when confronted with previously unseen inputs. We call the procedure `VecUQ-OT`.
4. We evaluate across domains (synthetic, images, and text) and downstream UQ tasks, specifically out-of-distribution detection, misclassification detection, selective prediction (images), and selective generation (text), and demonstrate robust performance of our composite measures.

## 2 METHOD

Most UQ pipelines use a *single* scalar as an uncertainty score, which forces practitioners to pick the most appropriate UQ measure for every dataset and task. We aim to avoid this early, all-in choice, postpone scalarization, and view different uncertainty measures as complementary signals of model confidence. Thus, we consider a *vector* of various uncertainty scores and impose a principled *ordering* of such vectors using ideas of Monge-Kantorovich ranks.

## 2.1 VECTORIZING UNCERTAINTY SIGNALS

Given an input $x \in \mathbb{R}^d$, we collect $m$ one-dimensional uncertainty measures (scores) into a vector

$$\mathbf{S}(x) = \begin{bmatrix} S_1(x) \\ \vdots \\ S_m(x) \end{bmatrix} \in \mathbb{R}_+^m, \tag{1}$$

where the uncertainty scores $S_j(x) \in \mathbb{R}_+$ may be of completely different nature (e.g., risk-based (Kotelevskii et al., 2025b; Schweighofer et al., 2025; Hofman et al., 2024), density-based (Lee et al., 2018; Mukhoti et al., 2023), etc.). Thus, $\mathbf{S}(x)$ combines heterogeneous information and allows us to take all signals into account simultaneously. We use *positive* uncertainty measures in our experiments. However, the approach also supports signed scores by choosing a symmetric (about the origin) reference distribution (Thurin et al., 2025; Chernozhukov et al., 2017).

## 2.2 ORDERING VECTORS VIA OT RANKS (VECUQ-OT)

We aim to compare *full* uncertainty vectors $\mathbf{S}(x)$ without committing fully to any of their components. For this, we place all uncertainty calibration vectors in a common, simple "reference space" using optimal transport. The uncertainty of these vectors is then defined as their distance from the reference center.

Let $\mathcal{D}_{\text{cal}} = \{x_i\}_{i=1}^n$ be a calibration set of *in-distribution* (ID) inputs, and let $\mathbf{s}_i = \mathbf{S}(x_i) \in \mathbb{R}_+^m$ be the corresponding uncertainty vectors. Consider the empirical distribution $\mu$ of $\{\mathbf{s}_i\}_{i=1}^n$ and some reference distribution $\nu$ on $\mathbb{R}_+^m$ (e.g., any isotropic distribution in $\mathbb{R}_+^d$; see Section 3). The reference distribution $\nu$ is represented via a discretization over $n$ target points $\{\tilde{\mathbf{s}}_j\}_{j=1}^n \subset \mathbb{R}_+^m$ with masses $\{\nu_j\}$. We consider a Monge-Kantorovich (MK) *rank map*, which is a measure-preserving transport $T \colon \mathbb{R}_+^m \to \mathbb{R}_+^m$ such that $T_{\#}\mu = \nu$ (Chernozhukov et al., 2017; Hallin et al., 2021; Hallin & Konen, 2024).

We define the *rank vector* of an uncertainty vector $\mathbf{S}(x)$ under the transport map $T$ as

$$R(x) = T\big(\mathbf{S}(x)\big),$$

yielding a center-outward ordering of uncertainty vectors via $r(x) = \|R(x)\|$. More precisely, for two inputs $x, x'$ we say that $\mathbf{S}(x)$ is more uncertain than $\mathbf{S}(x')$ if $r(x) > r(x')$.

Overall, VecUQ-OT transports uncertainty vectors to a canonical reference and defines resulting uncertainty as a distance to the reference center. In practice, $T$ can be approximated with computational OT (e.g., entropic OT) and evaluated via the barycentric projection.

**Entropic OT fit.** With cost $C_{ij} = \|\mathbf{s}_i - \tilde{\mathbf{s}}_j\|_2^2$, we fit an entropy-regularized OT *plan* $P^\epsilon$ between the empirical source $\{\mathbf{s}_i\}$ and the discrete target $\{\tilde{\mathbf{s}}_j\}$ by solving

$$\min_{P \in \Pi(\mu,\nu)} \langle C, P \rangle + \epsilon \sum_{i,j} P_{ij} \log P_{ij},$$

where $\Pi(\mu,\nu) = \{P \geq 0 \colon P\mathbf{1} = \mu, \ P^\top \mathbf{1} = \nu\}$. The minimizer has the Sinkhorn form

$$P^\epsilon = \text{diag}(u)\, K\, \text{diag}(v), \qquad K_{ij} = \exp\big(-C_{ij}/\epsilon\big),$$

with positive scalings $u, v$ chosen to match the marginals (Cuturi, 2013). Entropic OT is fast, numerically stable, and provides a smooth coupling between the two clouds.

**From plan to rank vector.** Rather than searching for a strict one-to-one map, we evaluate any point via the *barycentric projection* of the learned plan (Peyré et al., 2019). For a new input $x$ with $\mathbf{s} = \mathbf{S}(x)$, we introduce

the MK *rank image* of $x$:

$$\widehat{R}_\epsilon(x) = \sum_{j=1}^{n} \tilde{\mathbf{s}}_j \, w_j^\epsilon(\mathbf{s}), \quad \text{where} \quad w_j^\epsilon(\mathbf{s}) = \frac{v_j \exp\{-\|\mathbf{s} - \tilde{\mathbf{s}}_j\|_2^2/\epsilon\}}{\sum_{\ell=1}^{n} v_\ell \exp\{-\|\mathbf{s} - \tilde{\mathbf{s}}_\ell\|_2^2/\epsilon\}}. \tag{2}$$

The vector $\widehat{R}_\epsilon(x)$ determines the location of $\mathbf{S}(x)$ in the reference space once aligned by OT.

## 3 IMPLEMENTATION

### 3.1 LIMITATIONS OF BARYCENTRIC PROJECTION AND IMPLICATIONS FOR OUR METHOD

By construction (see equation (2)), the rank image $\widehat{R}_\epsilon(x)$ (the barycentric projection) is a convex combination of the *target* atoms $\{\tilde{\mathbf{s}}_j\}$, i.e.,

$$\widehat{R}_\epsilon(x) \in \text{conv}\{\tilde{\mathbf{s}}_j\}_{j=1}^{n}.$$

Consequently, once the coupling $P^\epsilon$ has been fit, the barycentric map cannot produce outputs outside the convex hull of the target support.

In our setting, the coupling is learned on *in-distribution* uncertainty vectors but must be applied to arbitrary inputs, including OOD ones. This means that mapped rank vectors for any new $x$ satisfy $\widehat{R}_\epsilon(x) \in \text{conv}\{\tilde{\mathbf{s}}_j\}_{\text{cal}}$. This leads to two undesirable effects:

1. *No extrapolation.* Inputs outside the calibration range cannot be mapped beyond the target hull; they are compressed toward it.

2. *Barycenter collapse far from support.* For $\mathbf{s}$ far from all $\tilde{\mathbf{s}}_j$, the kernel terms $\exp\{-c(\mathbf{s}, \tilde{\mathbf{s}}_j)/\epsilon\}$ are nearly equal, so $w_j^\epsilon(\mathbf{s}) \approx v_j / \sum_\ell v_\ell$ and $\widehat{R}_\epsilon(x) \to \bar{\mathbf{s}}_v := \sum_j v_j \tilde{\mathbf{s}}_j / \sum_j v_j$, a dataset-dependent barycenter. This reduces sensitivity to OOD structure.

These limitations motivate our design choices below, which mitigate the convex-hull restriction while retaining the computational benefits of entropic OT.

**Extending the source support: outer anchors.** The barycentric projection maps into $\text{conv}\{\tilde{\mathbf{s}}_j\}$, which can compress truly OOD *source* vectors toward the interior (see Section 3.1). To mitigate this while retaining entropic OT, we augment the *source* support with *outer anchor* points placed just beyond the calibration range. For each coordinate $k \in \{1, \ldots, m\}$, let $M_k = \max_i s_{i,k}$ and fix $\gamma > 1$ (we use $\gamma = 5$). Then we add

$$\mathcal{A} = \left\{ a \in \mathbb{R}_+^m: \ a_k \in \{0, \gamma M_k\} \text{ for all } k, \ a \neq 0 \right\},$$

the $2^m - 1$ nonzero corners of the box $[0, \gamma M_1] \times \cdots \times [0, \gamma M_m]$. This expands the *source* domain so that inputs with scores outside the calibration range are matched, under the learned coupling, to target points nearer the *boundary* of $\text{conv}\{\tilde{\mathbf{s}}_j\}$ rather than collapsing to its barycenter.

### 3.2 REFERENCE DISTRIBUTION AND DISCRETIZATION

We use two simple, isotropic factorized choices for the target $\nu$: (i) a product of exponentials (unbounded support on $\mathbb{R}_+^m$), and (ii) a product of beta distributions (bounded support on $[0, 1]^m$). Because we solve a *discrete* OT problem, we need target samples $\{\tilde{\mathbf{s}}_j\} \sim \nu$. To obtain them, we first draw a uniform grid in $[0, 1]^m$ and transform each coordinate using the inverse CDF of the chosen marginal (exponential or beta). Any strictly increasing radial reparameterization yields the same order, so we do not enforce normalization of rank vectors to the unit ball. Isotropy treats all coordinates symmetrically. Implementation details are in Appendix B.

## 3.3 COMPONENT SCALING

To improve the numerical stability of fitting entropy-regularized OT, we scale the components of $\mathbf{S}(x)$. We consider two options:

- **Global min-max scaling** (one scale for all components): preserves cross-component correlations.
- **Feature-wise min-max scaling** (per-component scales): may distort correlations but is more robust when component ranges differ greatly.

Our experiments showed that feature-wise scaling performs better, especially when measures have disparate ranges. An ablation study of design choices and a discussion of limitations are provided in Section C and Section D, respectively.

With all these design choices, we refer to the procedure as `VecUQ-OT`.

## 4 RELATED WORK

Our work lies at the intersection of two fields: uncertainty quantification (UQ) and optimal transport (OT). Consequently, we organize this section into two subsections, reviewing each area separately.

### 4.1 UNCERTAINTY QUANTIFICATION

In UQ, practitioners distinguish two primary sources of uncertainty: *aleatoric uncertainty* (AU) and *epistemic uncertainty* (EU) (Hüllermeier & Waegeman, 2021).

**Aleatoric uncertainty** reflects irreducible randomness in the data, a non-deterministic dependency between covariates $x$ and labels $y$, and is fully characterized by the ground-truth conditional distribution $p(y \mid x)$. Any divergence-based statistic (e.g., entropy, variance) of this distribution can serve as an AU measure (Kotelevskii et al., 2025b; Schweighofer et al., 2025; Hofman et al., 2024). Since the true $p(y \mid x)$ is unknown in practice, one approximates it via predictive models and estimates AU accordingly.

**Epistemic uncertainty** is considerably more challenging to define because it can manifest in diverse ways. Fundamentally, it reflects a lack of knowledge about the data-generating process. Accurate EU quantification is crucial in many downstream tasks such as OOD detection, active learning, and anomaly or novelty detection (see (Yang et al., 2024) and references therein), each of which emphasizes different aspects of uncertainty. As a result, these varied forms complicate the search for a single, unified definition of the EU.

A theoretically grounded definition of EU arises from statistical risk decomposition, where EU is the component of error unexplained by aleatoric (ground-truth) randomness (Kotelevskii et al., 2025b; 2022; Lahlou et al., 2024; Schweighofer et al., 2025; Hofman et al., 2024).

However, as discussed in (Kotelevskii et al., 2025b; Jiménez et al., 2025; Kotelevskii et al., 2024), $p(y \mid x)$, and hence AU, is only meaningful for in-distribution inputs. For OOD inputs (e.g., an animal image passed to an MNIST (Deng, 2012) classifier), $p(y \mid x)$ has no interpretable behavior. Consequently, this EU formulation is best suited to the "in-distribution" notion of EU, making it closely related to calibration (Guo et al., 2017; Ahdritz et al., 2025; Johnson et al., 2024; Kotelevskii et al., 2025a).

Another limitation of the risk-based definition of EU is evident for the OOD problem, a standard EU benchmark. OOD detection distinguishes between inputs $x$ seen during training and those outside the training distribution. Many practical UQ methods for OOD detection use **discriminative** models. The core assumption is that, for a Bayesian model or predictor ensemble, OOD inputs exhibit significant prediction disagreement, reflecting high EU (Kotelevskii et al., 2025b; Schweighofer et al., 2023; Houlsby et al., 2011; Hofman et al., 2024; Schweighofer et al., 2025).

Several points are worth emphasizing. First, OOD detection is more closely tied to estimating $p(x)$ than $p(y \mid x)$, and density-based measures have been successful for this problem (Kotelevskii et al., 2022; Lee et al., 2018; Mukhoti et al., 2023). Second, defining "disagreement" requires multiple models (Bayesian or ensembles), increasing computational cost. Third, such disagreement on OOD data is not induced "out of the box". However, one may encourage it, for example, via explicit entropy-maximization objectives (de Mathelin et al., 2025) and sufficiently large model sets.

To address these issues, uncertainty measures based on the density of neural network representations were introduced (Lee et al., 2018; Kotelevskii et al., 2022; Mukhoti et al., 2023). These do not require Bayesian models or ensembles, making them computationally cheaper.

In (Kotelevskii et al., 2022), a connection between density-based and risk-decomposition measures is shown for a specific loss and meta-model (a model trained on base-model embeddings), specifically Nadaraya-Watson kernel regression. Yet, risk- and density-based measures typically operate on different scales and arise from distinct frameworks, complicating their joint use. As we will see, our approach considers these measures jointly and yields a natural estimate of total uncertainty, given the different natures of the uncertainty components.

Lastly, the risk-decomposition framework assumes an **additive** split of uncertainties. While theoretically grounded, this assumption can break down in practice, where both AU and EU are estimated from the same model and, under limited data, may become highly interwoven (Wimmer et al., 2023). Moreover, as discussed above, AU is undefined for OOD inputs, so its values can be arbitrary. Consequently, any additive decomposition yields an equally arbitrary total-uncertainty (TU) estimate. By contrast, our multidimensional framework supports **non-additive** aggregation of AU and EU, potentially addressing the issues highlighted in (Wimmer et al., 2023).

### 4.2 OPTIMAL TRANSPORT AND MULTIVARIATE ORDERING

Optimal transport provides a geometric framework for comparing probability distributions by transporting mass from a source to a target at minimal cost (Peyré et al., 2019). Entropic regularization yields scalable solvers via Sinkhorn iterations (Cuturi, 2013).

Beyond distribution matching, recent work connects OT to multivariate quantiles and ranks, leading to vector orderings and measure-preserving maps in multiple dimensions (Chernozhukov et al., 2017; Hallin et al., 2021; Hallin & Konen, 2024). These constructions supply a principled notion of "order" for vectors and a distribution-aware scale against which multivariate observations can be compared.

OT-based orderings have enabled practical methods, including multidimensional conformal prediction (Thurin et al., 2025; Klein et al., 2025) and multivariate nonparametric testing (Ghosal & Sen, 2022).

## 5 EXPERIMENTS

This section evaluates our approach across tasks, datasets, and modalities. As described above and detailed in Section D, in all experiments we assume that no explicit OOD data are available when training OT, which is a realistic scenario in practice. We split the original validation set into two disjoint parts to fit optimal transport: a smaller subset for training the OT mapping (the calibration dataset) and the remaining "new" validation set for evaluation. Note that we require only covariates for this calibration dataset, *no labels are needed*. Unless stated otherwise, for `VecUQ-OT` we use a Beta target distribution, $\epsilon = 0.5$, and feature-wise min-max scaling for components.

Due to space constraints, additional experiments (e.g., analysis of extreme composition scenarios) are provided in Appendix E.

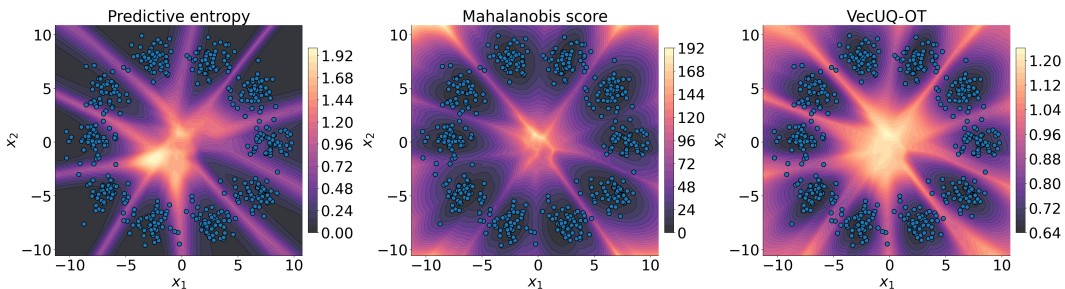

Figure 2: Resulting uncertainty measures on the toy dataset. *Left:* predictive entropy (aleatoric uncertainty). *Middle:* Mahalanobis score (epistemic proxy). *Right:* `VecUQ-OT` (our aggregated total uncertainty).

## 5.1 SYNTHETIC EXPERIMENTS

**Combining measures of different nature.** We begin with a two-dimensional synthetic classification problem with ten classes and a single deterministic model (a neural network with one hidden layer of 32 ReLU units). We estimate AU via a risk-based measure (predictive entropy). Since there is no ensemble or Bayesian model to measure EU directly, we use the Mahalanobis score (Lee et al., 2018) as an EU proxy. The dataset consists of ten Gaussian blobs (with standard deviation 1) uniformly placed on a circle of radius 8, each blob defining one class.

We then compute and combine the two uncertainty components as in equation (1). For each component,

$$\mathrm{AU}(x) = H\big[p_\theta(y \mid x)\big] = -\sum_{c=1}^{C} p_\theta(y = c \mid x) \log p_\theta(y = c \mid x),$$

and

$$\mathrm{EU}(x) = M(x) = \min_{c=1,\ldots,C} \big(f_\theta(x) - \hat{\mu}_c\big)^\top \hat{\Sigma}^{-1} \big(f_\theta(x) - \hat{\mu}_c\big),$$

where $\hat{\mu}_c = \frac{1}{N_c} \sum_{i:y_i=c} f_\theta(x_i)$ and $\hat{\Sigma} = \frac{1}{N} \sum_{c=1}^{C} \sum_{i:y_i=c} \big(f_\theta(x_i) - \hat{\mu}_c\big)\big(f_\theta(x_i) - \hat{\mu}_c\big)^\top$.

The two-dimensional uncertainty vector $(\mathrm{AU}, \mathrm{EU})$ jointly encodes aleatoric and epistemic uncertainty estimates, yielding a unified notion of *total uncertainty*. Applying our OT-based aggregation produces the map shown in the rightmost panel of Figure 2. Regions of high total uncertainty align with areas of label ambiguity (where predictive entropy peaks, left panel) and sparse coverage (where the Mahalanobis score is large, middle panel). This illustrates that our method effectively fuses the two uncertainty sources into a single, interpretable confidence score that highlights both class overlap and OOD regions.

## 5.2 IMAGE DATASETS

We study three downstream UQ problems: *out-of-distribution detection*, *misclassification detection*, and *selective prediction*. As base classifiers, following (Kotelevskii et al., 2025b), we train five independent deep-ensemble groups (ResNet18 (He et al., 2016)), each with four members (20 models total) with different random seeds (Lakshminarayanan et al., 2017). For risk-based uncertainties, we consider Logscore instantiations (Kotelevskii et al., 2025b; Schweighofer et al., 2025; Hofman et al., 2024). For `VecUQ-OT`, we use a Beta target distribution and feature-wise min–max scaling.

For all problems, as an example composite UQ measure we use a vector of four components: particular total, excess, and Bayes risk approximations combined with the Mahalanobis score. Other combinations are reported in Section F.

| In-Dist. | Out-of-Dist. | Ours | $R_{\text{exc}}^{1,1}$ | $R_{\text{tot}}^{1,1}$ | $R_{\text{b}}^1$ | MahS |
|---|---|---|---|---|---|---|
| CIFAR10 | CIFAR100 | **0.918 ± 0.001** | 0.905 ± 0.000 | 0.912 ± 0.001 | 0.917 ± 0.001 | 0.912 ± 0.002 |
| CIFAR10 | SVHN | 0.957 ± 0.005 | 0.943 ± 0.012 | 0.957 ± 0.007 | **0.963 ± 0.002** | 0.934 ± 0.005 |
| CIFAR10 | TinyImageNet | **0.912 ± 0.001** | 0.896 ± 0.000 | 0.904 ± 0.001 | 0.911 ± 0.001 | 0.910 ± 0.001 |
| CIFAR100 | CIFAR10 | 0.765 ± 0.001 | 0.725 ± 0.001 | **0.774 ± 0.002** | 0.773 ± 0.002 | 0.535 ± 0.004 |
| CIFAR100 | SVHN | 0.870 ± 0.006 | 0.756 ± 0.013 | 0.868 ± 0.006 | **0.870 ± 0.007** | 0.679 ± 0.034 |
| CIFAR100 | TinyImageNet | **1.000 ± 0.000** | **1.000 ± 0.000** | **1.000 ± 0.000** | 0.810 ± 0.001 | 0.623 ± 0.005 |
| TinyImageNet | ImageNet-A | **0.847 ± 0.002** | 0.781 ± 0.004 | 0.846 ± 0.002 | 0.835 ± 0.002 | 0.441 ± 0.009 |
| TinyImageNet | ImageNet-R | 0.835 ± 0.001 | 0.774 ± 0.003 | **0.837 ± 0.002** | 0.825 ± 0.003 | 0.405 ± 0.008 |
| TinyImageNet | ImageNet-O | **0.760 ± 0.002** | 0.753 ± 0.002 | 0.754 ± 0.002 | 0.724 ± 0.004 | 0.513 ± 0.005 |
| Avg | | **0.874 ± 0.002** | 0.837 ± 0.004 | 0.872 ± 0.002 | 0.848 ± 0.003 | 0.661 ± 0.008 |

Table 1: ROC-AUC of OOD detection: In-Dist./Out-of-Dist. pairs with composite and individual measures. Best (bold) and second-best (underlined) per row. Risk-based measures (notation follows (Kotelevskii et al., 2025b)) are instantiated by the Logscore proper scoring rule. MahS denotes the Mahalanobis score. "Ours" refers to VecUQ-OT.

| In-Dist. | Ours | $R_{\text{exc}}^{1,1}$ | $R_{\text{tot}}^{1,1}$ | $R_{\text{b}}^1$ | MahS |
|---|---|---|---|---|---|
| CIFAR10 | **0.944** ±.002 | 0.940 ±.003 | 0.943 ±.002 | 0.942 ±.002 | 0.928 ±.003 |
| CIFAR100 | 0.849 ±.003 | 0.818 ±.003 | **0.853** ±.003 | 0.845 ±.003 | 0.574 ±.006 |
| TinyImageNet | 0.847 ±.002 | 0.813 ±.001 | **0.851** ±.002 | 0.845 ±.003 | 0.417 ±.004 |
| Avg | 0.880 ±.002 | 0.857 ±.002 | **0.882** ±.002 | 0.877 ±.002 | 0.639 ±.004 |

(a) ROC-AUC of misclassification detection. Best (bold) and second-best (underlined) per row.

| In-Dist. | Ours | $R_{\text{exc}}^{1,1}$ | $R_{\text{tot}}^{1,1}$ | $R_{\text{b}}^1$ | MahS |
|---|---|---|---|---|---|
| CIFAR10 | **0.997** ±.000 | **0.997** ±.000 | **0.997** ±.000 | **0.997** ±.000 | 0.996 ±.000 |
| CIFAR100 | 0.916 ±.001 | 0.910 ±.001 | **0.918** ±.001 | 0.916 ±.001 | 0.811 ±.001 |
| TinyImageNet | 0.886 ±.001 | 0.879 ±.001 | **0.891** ±.000 | 0.889 ±.001 | 0.660 ±.005 |
| Avg | 0.933 ±.001 | 0.929 ±.001 | **0.935** ±.000 | 0.934 ±.001 | 0.822 ±.002 |

(b) Area under accuracy-coverage curve for selective prediction. Best (bold) and second-best (underlined) per row.

| Method | At Pareto Front (%) |
|---|---|
| Ours | **82.857%** |
| $R_{\text{tot}}^{1,1}$ | 65.714% |
| $R_{\text{b}}^1$ | 21.905% |
| $R_{\text{exc}}^{1,1}$ | 0.000% |
| MahS | 0.000% |

(c) Pareto dominance (percentage) across methods (higher is better).

**Out-of-distribution detection.** We evaluate on CIFAR10, CIFAR100 (Krizhevsky, 2009), and TinyImageNet (Le & Yang, 2015). For CIFAR10/100 as ID, OOD sets are CIFAR10, CIFAR100, TinyImageNet, and SVHN (Netzer et al., 2011). For TinyImageNet as ID, OOD sets are ImageNet-A, ImageNet-O (Hendrycks et al., 2021c), and ImageNet-R (Hendrycks et al., 2021a). We report ROC-AUC for ID vs. OOD ranking (higher is better); see Table 1.

Across datasets, our combined vector uncertainty is (i) never worse than the worst component in the vector, (ii) often the best single method, and (iii) best on average. Even when a component such as Mahalanobis underperforms, the combined score remains robust.

**Misclassification detection.** We use CIFAR10, CIFAR100, and TinyImageNet as IDs and report ROC-AUC for separating misclassified and correctly classified samples. Results are in Table 2a. VecUQ-OT is consistently robust, reaching the best or second-best score in all cases.

**Selective prediction.** For this problem, the metric is the *area under the accuracy–coverage curve*. We sort samples by uncertainty (ascending), add them individually, and compute accuracy at each coverage. This forms a curve, and the area under this curve is the score (higher is better). Results in Table 2b show that the combined measure is again robust across datasets.

**Pareto analysis across tasks.** Sometimes a component is best on a specific task, so improvements of a composition may not appear uniformly. To summarize performance across task pairs, we count how often a method lies on the Pareto front formed by the two corresponding metrics (we allow pairs of the same task type on different datasets). Table 2c shows that our method most often appears on the Pareto front, indicating strong overall robustness.

| Method | LLaMA-8B | | | Mistral-7B | | | Falcon-7B | | | Mean |
| --- | --- | --- | --- | --- | --- | --- | --- | --- | --- | --- |
| | Trivia | WMT19DeEn | MMLU | Trivia | WMT19DeEn | MMLU | Trivia | WMT19DeEn | MMLU | |
| Ours (Beta, FW) | 0.599 | **0.613** | 0.481 | **0.680** | **0.643** | 0.472 | 0.698 | **0.615** | 0.543 | **0.594** |
| Ours (Exp, FW) | 0.597 | 0.612 | 0.484 | 0.678 | 0.640 | 0.473 | 0.698 | 0.611 | 0.543 | 0.593 |
| CoCoA MSP | 0.603 | 0.584 | 0.492 | 0.677 | 0.607 | 0.469 | **0.699** | 0.590 | 0.539 | 0.584 |
| CoCoA PPL | 0.598 | 0.509 | 0.458 | 0.678 | 0.571 | 0.469 | 0.683 | 0.573 | 0.539 | 0.564 |
| CoCoA NMTE | 0.604 | 0.505 | 0.408 | 0.676 | 0.565 | 0.449 | 0.689 | 0.568 | 0.528 | 0.555 |
| MSP | 0.538 | 0.469 | **0.516** | 0.634 | 0.473 | **0.478** | 0.680 | 0.420 | **0.548** | 0.528 |
| PPL | 0.520 | 0.402 | 0.469 | 0.636 | 0.484 | **0.478** | 0.653 | 0.520 | 0.548 | 0.523 |
| Consistency | 0.615 | 0.450 | 0.395 | 0.652 | 0.499 | 0.427 | 0.657 | 0.487 | 0.493 | 0.520 |
| SAR | 0.593 | 0.472 | 0.360 | 0.638 | 0.519 | 0.418 | 0.647 | 0.503 | 0.520 | 0.519 |
| MTE | 0.506 | 0.391 | 0.362 | 0.623 | 0.477 | 0.459 | 0.642 | 0.532 | 0.543 | 0.504 |
| DegMat | **0.616** | 0.357 | 0.346 | 0.651 | 0.385 | 0.410 | 0.663 | 0.440 | 0.494 | 0.485 |
| EigValLaplacian | 0.600 | 0.283 | 0.296 | 0.622 | 0.330 | 0.401 | 0.656 | 0.400 | 0.471 | 0.451 |
| Semantic Entropy | 0.541 | 0.410 | 0.235 | 0.565 | 0.409 | 0.387 | 0.593 | 0.411 | 0.474 | 0.447 |
| Mean | 0.579 | 0.466 | 0.408 | 0.647 | 0.508 | 0.445 | 0.666 | 0.513 | 0.522 | 0.528 |

Table 3: Prediction Rejection Ratio (PRR; higher is better) for selective generation on text datasets. Columns group three models (LLaMA-8B, Mistral-7B, Falcon-7B) and three datasets each (Trivia, WMT19DeEn, MMLU); the last column reports the mean across all nine entries. The top block shows our OT-based combinations (`VecUQ-OT`); the middle block lists their individual components; the bottom block reports reference baselines. **Best** and second-best per column are highlighted.

## 5.3 TEXT DATASETS

We also consider experiments in the text domain, focusing on *selective generation*: using uncertainty as a proxy for quality and rejecting outputs accordingly. Dataset and baseline details appear in Section G.

We use the Prediction Rejection Ratio (PRR; Malinin & Gales, 2021) as the performance metric. PRR assesses how the average quality of generated outputs changes as an increasing percentage of outputs is rejected based on uncertainty. Formally, PRR is the ratio of the area between the Prediction-Rejection (PR) curve for a given uncertainty score and a random baseline, to the area between the ideal oracle (which perfectly ranks instances by quality) and the random baseline:

$$\text{PRR} = \frac{\text{AUC}_{\text{unc}} - \text{AUC}_{\text{rnd}}}{\text{AUC}_{\text{oracle}} - \text{AUC}_{\text{rnd}}}.$$

Results on textual data are reported in Table 3. Additional results are provided in Section G.

## 6 CONCLUSION

We studied a particular instance of inducing an order over uncertainty vectors via (unnormalized) Monge-Kantorovich ranks obtained from entropy-regularized discrete OT. To the best of our knowledge, this is the first attempt to formulate a general notion of ordering *multidimensional* uncertainty vectors. Other instantiations are possible, for example, amortized OT (Amos, 2023) or normalizing flows with cyclical-monotonicity constraints (Huang et al., 2021). However, both require training additional parametric models and thus introduce parameter-estimation uncertainty. In contrast, the entropy-regularized OT approach adopted here is nonparametric and avoids training a separate model.

Empirically, `VecUQ-OT` provides a robust, label-free calibration layer that aggregates heterogeneous UQ signals and performs competitively across domains and tasks without committing to a single scalar measure.

## USE OF LARGE LANGUAGE MODELS

We employed large language models (LLMs) as general-purpose assistants while preparing this manuscript. Their role was limited to two areas: (i) polishing wording to enhance clarity and readability, and (ii) coding support, suggesting completions and helping diagnose bugs. LLMs were not used for research ideation, experimental design, theoretical development, or interpretation of results. All substantive elements: problem formulation, methodology, and experiments, were conceived and executed solely by the authors.

## REPRODUCIBILITY STATEMENT

We provide the full code to reproduce our experiments as supplementary material and will release it publicly upon acceptance. All experiments were conducted on publicly available datasets or datasets we created ourselves, which will be released alongside the code. We ran experiments with multiple seeds, if applicable, and report summary statistics.

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

## A    MOTIVATING SYNTHETIC EXPERIMENT

Here, we provide details of a synthetic experiment that illustrates robustness when only one component works well for a given downstream task.

We consider the following binary classification problem. Two Gaussians are centered at $(-0.8, 0.0)$ and $(0.8, 0.2)$, sharing covariance $\begin{bmatrix} 1.0 & 0.6 \\ 0.6 & 1.2 \end{bmatrix}$. Samples from the first Gaussian are labeled 0, and samples from the second are labeled 1. Data are shown in Figure 3 (left). As a model, we use logistic regression.

We evaluate two downstream UQ problems: *misclassification detection* and *out-of-distribution (OOD) detection*.

We compare two uncertainty measures: the maximum softmax probability (MSP) (Hendrycks & Gimpel, 2017)—recoverable as the zero-one proper scoring-rule instantiation of the Bayes risk (Kotelevskii et al., 2025b)—and the Mahalanobis score (Lee et al., 2018).

The OOD data (another Gaussian) are placed so that the logistic regression is overly confident in assigning a class (Figure 3, right).

Accordingly, MSP performs well for misclassification detection but fails on OOD detection, while Mahalanobis behaves oppositely. This is exactly what we observe in Figure 1 (main text). Our proposed composite, `VecUQ-OT`, is robust across both tasks. Numerical results appear in Table 4.

Scalar measures are typically task-specific. By contrast, combining complementary signals with `VecUQ-OT` yields robustness for both misclassification and OOD detection (see Figure 1).

| Task | 1-MSP | Mahalanobis | VecUQ-OT |
|---|---|---|---|
| Misclassification detection | 0.766 | 0.445 | 0.710 |
| OOD detection | 0.099 | 1.000 | 1.000 |

Table 4: ROC-AUC on two downstream tasks: misclassification and OOD detection. Each individual measure (1-MSP or Mahalanobis) excels at only one task. Our vector-based combination (`VecUQ-OT`) remains robust across both.

## B    DETAILS ON SAMPLING FROM REFERENCE DISTRIBUTION

Our OT-based ordering maps calibration vectors $\mathbf{s}_i \in \mathbb{R}_+^m$ to a simple, factorized reference distribution $\nu$ and ranks points by radius in the reference space (via barycentric projection). Because we solve a *discrete* entropic OT problem, we approximate $\nu$ with a finite cloud of target points $\{\tilde{\mathbf{s}}_j\}_{j=1}^n$. This section details how we sample those targets.

We first draw points $U \in [0, 1]^{n \times m}$ in the unit hypercube using a Cartesian grid. Then, we transform each coordinate of $U$ with the inverse CDF of the chosen marginal, yielding independent coordinates and a factorized $\nu$:

- **Product of exponentials.** For rates $\lambda_\ell > 0$,

$$\tilde{\mathbf{s}}_{j\ell} = F_{\text{Exp}(\lambda_\ell)}^{-1}(U_{j\ell}) = -\frac{1}{\lambda_\ell} \log(1 - U_{j\ell}), \qquad \text{support } \mathbb{R}_+^m.$$

We use a common rate $\lambda_\ell \equiv \lambda$ for all coordinates.

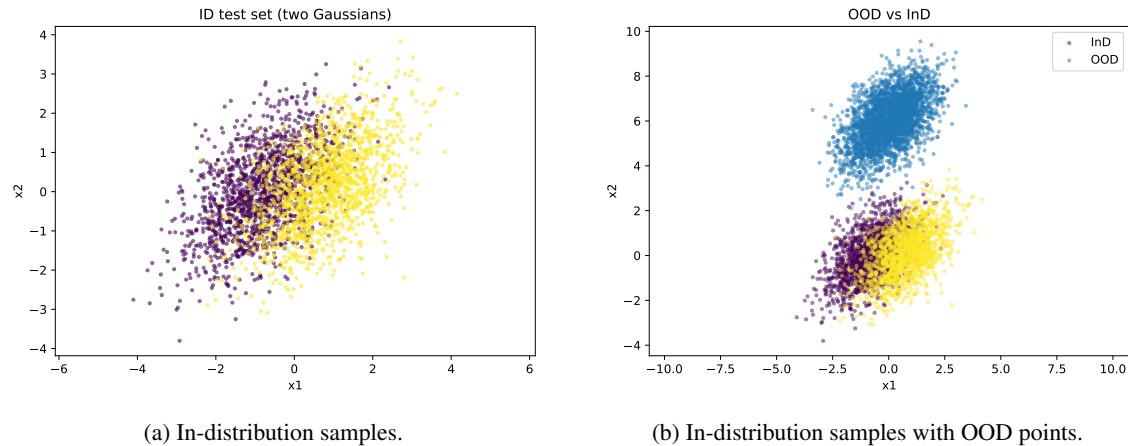

(a) In-distribution samples.     (b) In-distribution samples with OOD points.

Figure 3: Synthetic data used in the synthetic experiment.

- **Product of betas.** For $\alpha_\ell, \beta_\ell > 0$,

$$\tilde{\mathbf{s}}_{j\ell} = F^{-1}_{\text{Beta}(\alpha_\ell, \beta_\ell)}(U_{j\ell}), \qquad \text{support } [0,1]^m.$$

  Analogously, we take $\alpha_\ell = 1, \beta_\ell = 1$ for all coordinates. We use `scipy.special.betaincinv` for the inverse.

## C  ABLATION ON DESIGN CHOICES

We study how three design factors affect performance: (i) the *target distribution* for the OT reference cloud (Beta on $[0,1]^m$ vs. Exponential on $\mathbb{R}_+^m$), (ii) the *component scaling* (Feature-wise = per-feature min–max, Global = one min–max over all features, Identity = no scaling), and (iii) the *outer-anchor grid size* (GridSize $\in \{5, 2, 0\}$, where $0$ means no anchors).

We evaluate OOD detection on two ID/OOD pairs: CIFAR10 vs. CIFAR100 (Krizhevsky, 2009) and TinyImageNet (Le & Yang, 2015) vs. ImageNet-R (Hendrycks et al., 2021a). As a composite uncertainty vector, we use the combination from Section 5.2: $R_b^1, R_{\text{exc}}^{1,1}, R_{\text{tot}}^{1,1}, \text{MahS}$. Results are in Table 5; we use four ensemble groups of five members and report mean $\pm$ std ROC-AUC. From Table 5, results are close overall, with Feature-wise scaling typically strongest.

Next, we consider a different composition: various excess-risk approximations with log-score instantiation, $R_{\text{exc}}^{1,1}, R_{\text{exc}}^{1,2}, R_{\text{exc}}^{2,1}, R_{\text{exc}}^{1,3}, R_{\text{exc}}^{3,1}$. Results (Table 6) are again close overall; the Beta target with Feature-wise scaling performs best.

For TinyImageNet vs. ImageNet-R, we evaluate the same combinations. Results are in Tables 7 and 8. In Table 7, Identity scaling is omitted because the entropic-OT fit produced NaNs due to components with highly disparate magnitudes (e.g., MahS). We also observe that Global scaling performs poorly for this dataset/measure choice, whereas the Beta target with Feature-wise scaling remains robust. Overall, the observations are consistent with the previous ID/OOD pair.

| InD | OOD | ROC AUC | Target | ScalingType | GridSize |
|-----|-----|---------|--------|-------------|----------|
| CIFAR10 | CIFAR100 | $0.917984 \pm 0.000778$ | Beta | FeatureWise | 5 |
| CIFAR10 | CIFAR100 | $0.917983 \pm 0.000779$ | Beta | FeatureWise | 2 |
| CIFAR10 | CIFAR100 | $0.917950 \pm 0.000783$ | Beta | FeatureWise | 0 |
| CIFAR10 | CIFAR100 | $0.917474 \pm 0.000840$ | Exp | FeatureWise | 0 |
| CIFAR10 | CIFAR100 | $0.917423 \pm 0.000835$ | Exp | FeatureWise | 2 |
| CIFAR10 | CIFAR100 | $0.917394 \pm 0.000832$ | Exp | FeatureWise | 5 |
| CIFAR10 | CIFAR100 | $0.915669 \pm 0.000847$ | Exp | Identity | 5 |
| CIFAR10 | CIFAR100 | $0.915552 \pm 0.000814$ | Exp | Identity | 2 |
| CIFAR10 | CIFAR100 | $0.915245 \pm 0.001195$ | Exp | Identity | 0 |
| CIFAR10 | CIFAR100 | $0.913731 \pm 0.001273$ | Beta | Global | 5 |
| CIFAR10 | CIFAR100 | $0.913727 \pm 0.001273$ | Beta | Global | 2 |
| CIFAR10 | CIFAR100 | $0.913700 \pm 0.001274$ | Beta | Global | 0 |
| CIFAR10 | CIFAR100 | $0.913466 \pm 0.001299$ | Exp | Global | 5 |
| CIFAR10 | CIFAR100 | $0.913454 \pm 0.001300$ | Exp | Global | 2 |
| CIFAR10 | CIFAR100 | $0.913383 \pm 0.001302$ | Exp | Global | 0 |
| CIFAR10 | CIFAR100 | $0.910275 \pm 0.000686$ | Beta | Identity | 0 |
| CIFAR10 | CIFAR100 | $0.907669 \pm 0.000635$ | Beta | Identity | 2 |
| CIFAR10 | CIFAR100 | $0.907524 \pm 0.000649$ | Beta | Identity | 5 |

Table 5: Ablation on target distribution, scaling, and outer-anchor grid size for CIFAR10 vs. CIFAR100. Composite: $R_b^1, R_{\text{exc}}^{1,1}, R_{\text{tot}}^{1,1}, \text{MahS}$. Mean $\pm$ std ROC–AUC over four ensembles of five members.

| InD | OOD | ROC AUC | Target | ScalingType | GridSize |
|-----|-----|---------|--------|-------------|----------|
| CIFAR10 | CIFAR100 | $0.904907 \pm 0.000304$ | Beta | FeatureWise | 0 |
| CIFAR10 | CIFAR100 | $0.904833 \pm 0.000305$ | Beta | FeatureWise | 5 |
| CIFAR10 | CIFAR100 | $0.904826 \pm 0.000306$ | Beta | FeatureWise | 2 |
| CIFAR10 | CIFAR100 | $0.904776 \pm 0.000290$ | Beta | Identity | 0 |
| CIFAR10 | CIFAR100 | $0.904726 \pm 0.000288$ | Beta | Global | 0 |
| CIFAR10 | CIFAR100 | $0.904714 \pm 0.000295$ | Beta | Identity | 5 |
| CIFAR10 | CIFAR100 | $0.904710 \pm 0.000295$ | Beta | Identity | 2 |
| CIFAR10 | CIFAR100 | $0.904683 \pm 0.000292$ | Beta | Global | 5 |
| CIFAR10 | CIFAR100 | $0.904681 \pm 0.000292$ | Beta | Global | 2 |
| CIFAR10 | CIFAR100 | $0.904659 \pm 0.000298$ | Exp | FeatureWise | 0 |
| CIFAR10 | CIFAR100 | $0.904621 \pm 0.000313$ | Exp | FeatureWise | 2 |
| CIFAR10 | CIFAR100 | $0.904621 \pm 0.000313$ | Exp | FeatureWise | 5 |
| CIFAR10 | CIFAR100 | $0.904593 \pm 0.000289$ | Exp | Global | 0 |
| CIFAR10 | CIFAR100 | $0.904569 \pm 0.000296$ | Exp | Global | 2 |
| CIFAR10 | CIFAR100 | $0.904567 \pm 0.000296$ | Exp | Global | 5 |
| CIFAR10 | CIFAR100 | $0.904473 \pm 0.000278$ | Exp | Identity | 0 |
| CIFAR10 | CIFAR100 | $0.904417 \pm 0.000289$ | Exp | Identity | 5 |
| CIFAR10 | CIFAR100 | $0.904413 \pm 0.000288$ | Exp | Identity | 2 |

Table 6: Ablation with excess-risk variants for CIFAR10 vs. CIFAR100. Composite: $R_{\text{exc}}^{1,1}, R_{\text{exc}}^{1,2}, R_{\text{exc}}^{2,1}, R_{\text{exc}}^{1,3}, R_{\text{exc}}^{3,1}$. Mean $\pm$ std ROC–AUC over four ensembles of five members.

## D  LIMITATIONS

We highlight several limitations of our approach.

1. **Need for a separate calibration set.** Although the OT-based mapping is lightweight to fit, it requires a separate *covariates-only* dataset for calibration. Ideally, this set would include both in-distribution and OOD samples, allowing us to observe uncertainty vectors where OOD-typical scores lie and avoiding the outer-anchor heuristic. However, relying on explicit OOD data reduces generality. Consequently, in our experiments, we assume no OOD data and instead calibrate on in-distribution samples, augmenting with synthetic anchor points in score space.

| InD | OOD | ROC AUC | Target | ScalingType | GridSize |
|-----|-----|---------|--------|-------------|----------|
| TinyImageNet | ImageNet-R | $0.834519 \pm 0.000639$ | Beta | FeatureWise | 5 |
| TinyImageNet | ImageNet-R | $0.834514 \pm 0.000639$ | Beta | FeatureWise | 2 |
| TinyImageNet | ImageNet-R | $0.834509 \pm 0.000652$ | Beta | FeatureWise | 0 |
| TinyImageNet | ImageNet-R | $0.831367 \pm 0.000777$ | Exp | FeatureWise | 0 |
| TinyImageNet | ImageNet-R | $0.831306 \pm 0.000835$ | Exp | FeatureWise | 2 |
| TinyImageNet | ImageNet-R | $0.831289 \pm 0.000829$ | Exp | FeatureWise | 5 |
| TinyImageNet | ImageNet-R | $0.421999 \pm 0.006611$ | Exp | Global | 5 |
| TinyImageNet | ImageNet-R | $0.421810 \pm 0.006610$ | Exp | Global | 2 |
| TinyImageNet | ImageNet-R | $0.420473 \pm 0.006612$ | Exp | Global | 0 |
| TinyImageNet | ImageNet-R | $0.419511 \pm 0.006852$ | Beta | Global | 5 |
| TinyImageNet | ImageNet-R | $0.419462 \pm 0.006853$ | Beta | Global | 2 |
| TinyImageNet | ImageNet-R | $0.419213 \pm 0.006850$ | Beta | Global | 0 |

Table 7: Ablation on target distribution, scaling, and outer-anchor grid size for TinyImageNet vs. ImageNet-R. Composite: $R_b^1, R_{\mathrm{exc}}^{1,1}, R_{\mathrm{tot}}^{1,1}, \mathrm{MahS}$. Mean $\pm$ std ROC–AUC over four ensembles of five members.

| InD | OOD | ROC AUC | Target | ScalingType | GridSize |
|-----|-----|---------|--------|-------------|----------|
| TinyImageNet | ImageNet-R | $0.777316 \pm 0.003418$ | Beta | FeatureWise | 5 |
| TinyImageNet | ImageNet-R | $0.777304 \pm 0.003420$ | Beta | FeatureWise | 2 |
| TinyImageNet | ImageNet-R | $0.777266 \pm 0.003377$ | Beta | FeatureWise | 0 |
| TinyImageNet | ImageNet-R | $0.776733 \pm 0.003649$ | Exp | FeatureWise | 2 |
| TinyImageNet | ImageNet-R | $0.776686 \pm 0.003647$ | Exp | FeatureWise | 5 |
| TinyImageNet | ImageNet-R | $0.776372 \pm 0.003673$ | Exp | FeatureWise | 0 |
| TinyImageNet | ImageNet-R | $0.774565 \pm 0.002815$ | Beta | Identity | 0 |
| TinyImageNet | ImageNet-R | $0.774534 \pm 0.002825$ | Beta | Identity | 5 |
| TinyImageNet | ImageNet-R | $0.774532 \pm 0.002832$ | Beta | Identity | 2 |
| TinyImageNet | ImageNet-R | $0.774266 \pm 0.002822$ | Beta | Global | 5 |
| TinyImageNet | ImageNet-R | $0.774248 \pm 0.002823$ | Beta | Global | 2 |
| TinyImageNet | ImageNet-R | $0.774174 \pm 0.002803$ | Beta | Global | 0 |
| TinyImageNet | ImageNet-R | $0.774099 \pm 0.002938$ | Exp | Global | 2 |
| TinyImageNet | ImageNet-R | $0.774089 \pm 0.002924$ | Exp | Global | 5 |
| TinyImageNet | ImageNet-R | $0.773737 \pm 0.002870$ | Exp | Global | 0 |
| TinyImageNet | ImageNet-R | $0.773190 \pm 0.002905$ | Exp | Identity | 2 |
| TinyImageNet | ImageNet-R | $0.773148 \pm 0.002895$ | Exp | Identity | 5 |
| TinyImageNet | ImageNet-R | $0.772700 \pm 0.002874$ | Exp | Identity | 0 |

Table 8: Ablation with excess-risk variants for TinyImageNet vs. ImageNet-R. Composite: $R_{\mathrm{exc}}^{1,1}, R_{\mathrm{exc}}^{1,2}, R_{\mathrm{exc}}^{2,1}, R_{\mathrm{exc}}^{1,3}, R_{\mathrm{exc}}^{3,1}$. Mean $\pm$ std ROC–AUC over four ensembles of five members.

2. **Choice of cost function.** We use squared Euclidean distance as the OT cost. While effective empirically, it may not be optimal for transporting one uncertainty representation to another. Exploring alternatives within entropy-regularized, discrete OT is nontrivial and may complicate convergence and stability.

3. **Radial ranking discards direction.** Our final scalar score depends only on the norm of the barycentric image (a radial rank), not its direction. Although the isotropic reference is designed to make angles uninformative, directional information can still carry task-specific signals that our current rank ignores.

## E   EXTREME COMPOSITIONS

We examine edge cases to test the robustness and sanity of our vector-based aggregation. In each case, `VecUQ-OT` should behave predictably: it must *not* invent structure when none is present and should preserve useful ordering when a signal exists. Results in this section are consistent across the OT design choices discussed in the main text (reference distribution, component normalization, etc.).

### E.1 SINGLE-COMPONENT VECTOR

We set the composite vector to contain a *single* uncertainty measure as a sanity check. In this setting, VecUQ-OT should reproduce the original ordering exactly, since there is no multivariate structure to exploit—this is what we observe. For example, Table 9 reports OOD ROC-AUC on CIFAR10 (ID) vs. CIFAR100 (OOD) using an excess-risk estimator (log-score instantiation (Kotelevskii et al., 2025b)). The ordering and ROC-AUC are unchanged after applying VecUQ-OT.

| In-distribution | Out-of-distribution | Measure | Mean ROC AUC | Std ROC AUC |
|---|---|---|---|---|
| CIFAR10 | CIFAR100 | $R_{exc}^{1,1}$ | 0.9047 | 0.0003 |
| CIFAR10 | CIFAR100 | VecUQ-OT | 0.9047 | 0.0003 |

Table 9: Single-component composition: the vector contains only $R_{exc}^{1,1}$. VecUQ-OT preserves the original ordering (identical ROC-AUC). Ensembles: 4 groups of 5 members.

### E.2 STACKING IDENTICAL MEASURES

Next, we stack the *same* measure multiple times (duplicate coordinates). The vector carries no additional information beyond the single measure, so the induced order should match the base measure again. Table 10 confirms this.

| In-distribution | Out-of-distribution | Measure | Mean ROC AUC | Std ROC AUC |
|---|---|---|---|---|
| CIFAR10 | CIFAR100 | $R_{exc}^{1,1}$ | 0.9047 | 0.0003 |
| CIFAR10 | CIFAR100 | $R_{exc}^{1,1}$ | 0.9047 | 0.0003 |
| CIFAR10 | CIFAR100 | $R_{exc}^{1,1}$ | 0.9047 | 0.0003 |
| CIFAR10 | CIFAR100 | $R_{exc}^{1,1}$ | 0.9047 | 0.0003 |
| CIFAR10 | CIFAR100 | $R_{exc}^{1,1}$ | 0.9047 | 0.0003 |
| CIFAR10 | CIFAR100 | VecUQ-OT | 0.9047 | 0.0003 |

Table 10: Duplicated-measure composition: the vector repeats $R_{exc}^{1,1}$ across coordinates. VecUQ-OT leaves the ranking unchanged, as expected. Ensembles: 4 groups of 5 members.

### E.3 ONE SIGNAL, THE REST CONSTANTS

Finally, we build a vector where *one* coordinate carries the signal and all others are constants. A sensible aggregator should ignore the constant coordinates and follow the informative ones. Table 11 shows that VecUQ-OT matches the ROC-AUC of the informative measure and is unaffected by constant dimensions.

| In-distribution | Out-of-distribution | Metric | Mean ROC AUC | Std ROC AUC |
|---|---|---|---|---|
| CIFAR10 | CIFAR100 | $R_{exc}^{1,1}$ | 0.9047 | 0.0003 |
| CIFAR10 | CIFAR100 | Constant 1 | 0.5000 | 0.0000 |
| CIFAR10 | CIFAR100 | Constant 2 | 0.5000 | 0.0000 |
| CIFAR10 | CIFAR100 | Constant 3 | 0.5000 | 0.0000 |
| CIFAR10 | CIFAR100 | Constant 4 | 0.5000 | 0.0000 |
| CIFAR10 | CIFAR100 | Constant 5 | 0.5000 | 0.0000 |
| CIFAR10 | CIFAR100 | VecUQ-OT | 0.9047 | 0.0003 |

Table 11: One-signal composition: the vector contains $R_{exc}^{1,1}$ plus several constant coordinates. VecUQ-OT follows the informative coordinate and ignores constants. Ensembles: 4 groups of 5 members.

# F  OTHER MEASURE CHOICES

In this section, we examine additional combinations of uncertainty measures.

**Aggregation of different total-uncertainty estimates.** *Total* uncertainty is typically decomposed into aleatoric and epistemic parts, aims to capture all sources of predictive uncertainty, and is often a safe default for downstream tasks. However, as shown in (Kotelevskii et al., 2025b; Schweighofer et al., 2025; Hofman et al., 2024), there are multiple valid instantiations via proper scoring rules and several Bayesian approximation strategies for the total uncertainty estimate. Here we fix one Bayesian estimate, $R_{\text{tot}}^{1,2}$, and follow the setup from Section 5.2 with four deep-ensemble groups of five members each. We then instantiate the total-risk estimate with several proper scoring rules (Logscore, Brier, Spherical, and 0-1) and form a composite vector from these components. Table 12 shows that `VecUQ-OT` is robust: on average, it performs best across problems and, in most cases, ranks best or second-best. Among individual measures, the Logscore instantiation often performs best, but it can occasionally underperform (e.g., CIFAR100 and TinyImageNet for misclassification detection and selective prediction), whereas `VecUQ-OT` remains stable.

| InD | Eval | VecUQ-OT | Logscore | Brier | Spherical | Zero-one |
|---|---|---|---|---|---|---|
| CIFAR10 | CIFAR10 [miscls] | **0.946 ± 0.002** | 0.945 ± 0.002 | **0.946 ± 0.002** | **0.946 ± 0.002** | **0.946 ± 0.002** |
| | CIFAR10 [selective] | **0.997 ± 0.000** | **0.997 ± 0.000** | **0.997 ± 0.000** | **0.997 ± 0.000** | **0.997 ± 0.000** |
| | CIFAR100 [ood] | 0.913 ± 0.001 | **0.916 ± 0.001** | 0.913 ± 0.001 | 0.913 ± 0.001 | 0.911 ± 0.001 |
| | SVHN [ood] | 0.958 ± 0.003 | **0.963 ± 0.003** | 0.958 ± 0.004 | 0.958 ± 0.004 | 0.955 ± 0.004 |
| | TinyImageNet [ood] | 0.906 ± 0.001 | **0.910 ± 0.001** | 0.906 ± 0.001 | 0.906 ± 0.001 | 0.904 ± 0.001 |
| CIFAR100 | CIFAR10 [ood] | 0.774 ± 0.002 | **0.775 ± 0.002** | 0.774 ± 0.002 | 0.774 ± 0.002 | 0.771 ± 0.001 |
| | CIFAR100 [miscls] | 0.864 ± 0.003 | 0.850 ± 0.003 | 0.866 ± 0.003 | 0.866 ± 0.003 | **0.870 ± 0.003** |
| | CIFAR100 [selective] | 0.922 ± 0.001 | 0.918 ± 0.001 | 0.922 ± 0.001 | 0.922 ± 0.001 | **0.924 ± 0.001** |
| | SVHN [ood] | 0.859 ± 0.006 | **0.870 ± 0.006** | 0.857 ± 0.006 | 0.857 ± 0.006 | 0.847 ± 0.006 |
| | TinyImageNet [ood] | 0.938 ± 0.001 | 0.926 ± 0.001 | 0.937 ± 0.001 | 0.937 ± 0.001 | **0.947 ± 0.002** |
| TinyImageNet | TinyImageNet [miscls] | 0.863 ± 0.003 | 0.850 ± 0.003 | 0.863 ± 0.003 | 0.863 ± 0.003 | **0.868 ± 0.004** |
| | TinyImageNet [selective] | 0.895 ± 0.001 | 0.891 ± 0.000 | 0.896 ± 0.001 | 0.896 ± 0.001 | **0.897 ± 0.001** |
| | ImageNet-A [ood] | 0.836 ± 0.003 | **0.841 ± 0.002** | 0.834 ± 0.003 | 0.834 ± 0.003 | 0.827 ± 0.003 |
| | ImageNet-R [ood] | 0.825 ± 0.003 | **0.831 ± 0.003** | 0.824 ± 0.003 | 0.824 ± 0.003 | 0.817 ± 0.003 |
| | ImageNet-O [ood] | **0.736 ± 0.003** | 0.735 ± 0.003 | **0.736 ± 0.003** | **0.736 ± 0.003** | 0.735 ± 0.004 |
| Mean | | 0.8821 ± 0.002 | 0.8811 ± 0.002 | 0.8819 ± 0.002 | 0.8819 ± 0.002 | 0.8809 ± 0.002 |

Table 12: Performance for compositions of different approximations of $R_{\text{tot}}^{1,2}$. We consider all problems from Sec. 5.2. Here, [miscls] denotes misclassification detection, [selective] selective prediction, and [ood] OOD detection. For [miscls] and [ood] we report ROC–AUC; for [selective] we report area under the accuracy–coverage curve.

**Aggregation of different aleatoric-uncertainty estimates.** *Aleatoric* uncertainty is used to detect regions with increased label noise. As shown in (Kotelevskii et al., 2025b; Schweighofer et al., 2025), it is also effective for OOD detection. Table 13 reports results for aggregating different $R_b^1$ estimates. The pattern mirrors Table 12: the Logscore-based measure most frequently attains the top score (and here is best on average), but for some problems it drops to the bottom. In contrast, our composite measure is very robust—never the worst and typically best or second-best.

**Aggregation of different epistemic-uncertainty estimates.** *Epistemic* uncertainty targets inputs where the model lacks sufficient knowledge of the data-generating process, which includes OOD detection. In this experiment, we aggregate several instantiations (induced by different proper scoring rules) of the excess-risk measure $R_{\text{exc}}^{1,3}$. Results are shown in Table 14. Consistent with previous observations, `VecUQ-OT` is highly robust. Logscore and Spherical perform well on average, but each is sometimes near the bottom, whereas `VecUQ-OT` is typically best or second-best.

| InD | Eval | VecUQ-OT | Logscore | Brier | Spherical | Zero-one |
|---|---|---|---|---|---|---|
| CIFAR10 | CIFAR10 [miscls] | **0.942 ± 0.002** | **0.942 ± 0.002** | **0.942 ± 0.002** | **0.942 ± 0.002** | **0.942 ± 0.002** |
| | CIFAR10 [selective] | **0.997 ± 0.000** | **0.997 ± 0.000** | **0.997 ± 0.000** | **0.997 ± 0.000** | **0.997 ± 0.000** |
| | CIFAR100 [ood] | 0.915 ± 0.001 | **0.917 ± 0.001** | 0.914 ± 0.001 | 0.915 ± 0.001 | 0.913 ± 0.001 |
| | SVHN [ood] | 0.959 ± 0.002 | **0.963 ± 0.002** | 0.958 ± 0.002 | 0.959 ± 0.002 | 0.956 ± 0.002 |
| | TinyImageNet [ood] | 0.909 ± 0.001 | **0.911 ± 0.001** | 0.909 ± 0.001 | 0.909 ± 0.001 | 0.907 ± 0.001 |
| CIFAR100 | CIFAR10 [ood] | **0.773 ± 0.002** | **0.773 ± 0.002** | **0.773 ± 0.002** | **0.773 ± 0.002** | 0.772 ± 0.002 |
| | CIFAR100 [miscls] | 0.855 ± 0.003 | 0.845 ± 0.003 | 0.858 ± 0.003 | 0.856 ± 0.003 | **0.859 ± 0.003** |
| | CIFAR100 [selective] | 0.919 ± 0.001 | 0.916 ± 0.001 | 0.920 ± 0.001 | 0.920 ± 0.001 | **0.921 ± 0.001** |
| | SVHN [ood] | 0.862 ± 0.006 | **0.870 ± 0.007** | 0.858 ± 0.006 | 0.861 ± 0.006 | 0.856 ± 0.006 |
| | TinyImageNet [ood] | 0.803 ± 0.001 | **0.810 ± 0.001** | 0.806 ± 0.001 | 0.806 ± 0.001 | 0.803 ± 0.001 |
| TinyImageNet | TinyImageNet [miscls] | 0.853 ± 0.003 | 0.845 ± 0.003 | **0.855 ± 0.003** | 0.853 ± 0.003 | **0.855 ± 0.003** |
| | TinyImageNet [selective] | 0.892 ± 0.001 | 0.889 ± 0.001 | **0.893 ± 0.001** | 0.892 ± 0.001 | **0.893 ± 0.001** |
| | ImageNet-A [ood] | 0.831 ± 0.003 | **0.835 ± 0.002** | 0.827 ± 0.003 | 0.830 ± 0.003 | 0.826 ± 0.003 |
| | ImageNet-R [ood] | 0.820 ± 0.003 | **0.825 ± 0.003** | 0.816 ± 0.003 | 0.819 ± 0.003 | 0.815 ± 0.003 |
| | ImageNet-O [ood] | 0.723 ± 0.004 | **0.724 ± 0.004** | 0.721 ± 0.004 | 0.723 ± 0.004 | 0.721 ± 0.004 |
| AVG | [all rows] | 0.870 ± 0.002 | 0.871 ± 0.002 | 0.869 ± 0.002 | 0.870 ± 0.002 | 0.869 ± 0.002 |

Table 13: Performance for compositions of different approximations of $R_b^1$. We consider all problems from Sec. 5.2. Here, [miscls] denotes misclassification detection, [selective] selective prediction, and [ood] OOD detection. For [miscls] and [ood] we report ROC–AUC; for [selective] we report area under the accuracy–coverage curve.

| InD | Eval | VecUQ-OT | Logscore | Brier | Spherical | Zero-one |
|---|---|---|---|---|---|---|
| CIFAR10 | CIFAR10 [miscls] | **0.942 ± 0.003** | 0.936 ± 0.003 | **0.942 ± 0.003** | **0.942 ± 0.003** | 0.797 ± 0.008 |
| | CIFAR10 [selective] | **0.997 ± 0.000** | **0.997 ± 0.000** | **0.997 ± 0.000** | **0.997 ± 0.000** | 0.983 ± 0.002 |
| | CIFAR100 [ood] | **0.904 ± 0.000** | 0.903 ± 0.001 | 0.902 ± 0.000 | **0.904 ± 0.000** | 0.755 ± 0.001 |
| | SVHN [ood] | 0.941 ± 0.011 | 0.940 ± 0.013 | 0.940 ± 0.010 | **0.942 ± 0.010** | 0.825 ± 0.038 |
| | TinyImageNet [ood] | **0.895 ± 0.000** | 0.894 ± 0.001 | 0.893 ± 0.001 | 0.894 ± 0.001 | 0.752 ± 0.002 |
| CIFAR100 | CIFAR10 [ood] | 0.710 ± 0.002 | **0.718 ± 0.002** | 0.681 ± 0.002 | 0.715 ± 0.002 | 0.689 ± 0.002 |
| | CIFAR100 [miscls] | 0.819 ± 0.003 | 0.804 ± 0.003 | 0.783 ± 0.005 | **0.827 ± 0.003** | 0.806 ± 0.003 |
| | CIFAR100 [selective] | 0.910 ± 0.001 | 0.906 ± 0.001 | 0.900 ± 0.002 | **0.914 ± 0.001** | 0.879 ± 0.003 |
| | SVHN [ood] | 0.718 ± 0.011 | **0.750 ± 0.016** | 0.662 ± 0.007 | 0.719 ± 0.010 | 0.706 ± 0.011 |
| | TinyImageNet [ood] | 0.995 ± 0.001 | **1.000 ± 0.000** | 0.953 ± 0.003 | 0.990 ± 0.001 | 0.976 ± 0.002 |
| TinyImageNet | TinyImageNet [miscls] | 0.799 ± 0.002 | 0.791 ± 0.002 | 0.754 ± 0.003 | **0.805 ± 0.001** | 0.801 ± 0.004 |
| | TinyImageNet [selective] | 0.875 ± 0.001 | 0.872 ± 0.001 | 0.860 ± 0.002 | **0.878 ± 0.001** | 0.853 ± 0.003 |
| | ImageNet-A [ood] | 0.716 ± 0.003 | **0.772 ± 0.004** | 0.651 ± 0.005 | 0.715 ± 0.002 | 0.713 ± 0.002 |
| | ImageNet-R [ood] | 0.715 ± 0.002 | **0.764 ± 0.003** | 0.657 ± 0.005 | 0.716 ± 0.004 | 0.712 ± 0.003 |
| | ImageNet-O [ood] | 0.721 ± 0.003 | **0.746 ± 0.003** | 0.691 ± 0.005 | 0.719 ± 0.003 | 0.700 ± 0.004 |
| AVG | [all rows] | 0.844 ± 0.003 | 0.853 ± 0.003 | 0.818 ± 0.003 | 0.845 ± 0.003 | 0.796 ± 0.006 |

Table 14: Performance for compositions of different approximations of $R_{\mathrm{exc}}^{1,3}$. We consider all problems from Sec. 5.2. Here, [miscls] denotes misclassification detection, [selective] selective prediction, and [ood] OOD detection. For [miscls] and [ood] we report ROC–AUC; for [selective] we report area under the accuracy–coverage curve.

## G  TEXTUAL EXPERIMENTS DETAILS

**Datasets.** We use datasets covering several NLP tasks in our experimental setup, including summarization, translation, and long- and short-form question answering. For summarization, we use the XSum dataset (Narayan et al., 2018). We use the WMT14 Fr-En and WMT19 De-En datasets (Bojar et al., 2014; Barrault et al., 2019) for translation. For short-form question answering, we use the MMLU and Gsm8k datasets (Hendrycks et al., 2021b; Cobbe et al., 2021), while for long-form question answering, we employ TriviaQA and CoQA (Joshi et al., 2017; Reddy et al., 2019).

| Method | CoQA | GSM8k | MMLU | Trivia | WMT14FrEn | WMT19DeEn | XSum | mean |
|---|---|---|---|---|---|---|---|---|
| Ours (Exp, FW) | 0.393 | 0.577 | 0.473 | 0.678 | 0.445 | 0.640 | 0.337 | 0.506 |
| Ours (Beta, FW) | 0.395 | 0.568 | 0.472 | 0.680 | 0.441 | 0.643 | 0.339 | 0.505 |
| CoCoA MSP | 0.403 | 0.548 | 0.469 | 0.677 | 0.402 | 0.607 | 0.331 | 0.491 |
| CoCoA PPL | 0.389 | 0.459 | 0.469 | 0.678 | 0.396 | 0.571 | 0.309 | 0.467 |
| CoCoA MTE | 0.376 | 0.490 | 0.449 | 0.676 | 0.397 | 0.565 | 0.312 | 0.466 |
| MSP | 0.350 | 0.491 | 0.478 | 0.634 | 0.332 | 0.473 | 0.278 | 0.434 |
| SAR | 0.332 | 0.483 | 0.418 | 0.638 | 0.372 | 0.519 | 0.089 | 0.407 |
| Consistency | 0.403 | 0.465 | 0.427 | 0.652 | 0.306 | 0.499 | 0.063 | 0.402 |
| PPL | 0.289 | 0.280 | 0.478 | 0.636 | 0.370 | 0.484 | 0.226 | 0.395 |
| MTE | 0.252 | 0.326 | 0.459 | 0.623 | 0.397 | 0.477 | 0.203 | 0.391 |
| DegMat | 0.352 | 0.327 | 0.410 | 0.651 | 0.224 | 0.385 | 0.130 | 0.354 |
| Semantic Entropy | 0.294 | 0.483 | 0.387 | 0.565 | 0.274 | 0.409 | -0.007 | 0.343 |
| EigValLaplacian | 0.316 | 0.264 | 0.401 | 0.622 | 0.201 | 0.330 | 0.126 | 0.323 |

Table 15: Mistral-7B, PRR (higher is better).

**Models.** For our experiments, we use the base versions of LLaMA-3.1 8B, Mistral-7B, and Falcon-7B (Dubey et al., 2024; Jiang et al., 2023; Team, 2024).

**Evaluation.** We solve the task of selective generation by using the uncertainty score as a proxy for quality and rejecting outputs based on it. However, traditional metrics like ROC-AUC or ECE are inapplicable for textual outputs. Not only do some uncertainty-estimation methods produce unbounded values, but generation quality is also non-discrete. Thus, we use the Prediction Rejection Ratio (PRR) (Malinin & Gales, 2021). PRR assesses how the average quality of generated outputs changes as an increasing percentage of these outputs is rejected based on the uncertainty scores. Formally,

$$PRR = \frac{\text{AUC}_{\text{unc}} - \text{AUC}_{\text{rnd}}}{\text{AUC}_{\text{oracle}} - \text{AUC}_{\text{rnd}}}. \tag{3}$$

Rejecting 100% of the outputs is impractical, so we evaluate PRR at a 50% rejection rate, indicating how effectively the uncertainty score identifies and rejects the least desirable generations.

**Quality Metrics.** For each evaluated task, we select a specific quality measure. For summarization, quality is assessed using the Align Score between input text and the output (Zha et al., 2023). For translation, we employ COMET (Rei et al., 2022). For short-form QA, we use Accuracy, while for long-form QA, we rely on the Align Score between the target answer and the generated output.

**Baselines.** We employ three simple token-probabilities-based baselines: Maximum Sequence Probability, Perplexity, and Mean Token Entropy (Fomicheva et al., 2020). Additionally, we use state-of-the-art methods: Semantic Entropy (Kuhn et al., 2023), SAR (Duan et al., 2024), Degree Matrix and Eigenvalue of the Graph Laplacian (Lin et al., 2023). We also include CoCoA variants and the Consistency score as defined in (Vashurin et al., 2025).

**Additional results.** We extend the results from the main text and provide additional datasets in Tables 15, 16, and 17.

| Method | CoQA | GSM8k | MMLU | Trivia | WMT14FrEn | WMT19DeEn | XSum | mean |
|---|---|---|---|---|---|---|---|---|
| Ours (Exp, FW) | 0.354 | 0.420 | 0.484 | 0.597 | 0.476 | 0.612 | 0.405 | 0.478 |
| Ours (Beta, FW) | 0.356 | 0.409 | 0.481 | 0.599 | 0.476 | 0.613 | 0.410 | 0.478 |
| CoCoA MSP | 0.367 | 0.362 | 0.492 | 0.603 | 0.443 | 0.584 | 0.395 | 0.464 |
| CoCoA PPL | 0.352 | 0.417 | 0.458 | 0.598 | 0.440 | 0.509 | 0.401 | 0.453 |
| CoCoA MTE | 0.351 | 0.433 | 0.408 | 0.604 | 0.436 | 0.505 | 0.392 | 0.447 |
| MSP | 0.290 | 0.310 | 0.516 | 0.538 | 0.320 | 0.469 | 0.341 | 0.398 |
| PPL | 0.261 | 0.284 | 0.469 | 0.520 | 0.346 | 0.402 | 0.383 | 0.381 |
| SAR | 0.324 | 0.389 | 0.360 | 0.593 | 0.412 | 0.472 | 0.057 | 0.372 |
| Consistency | 0.391 | 0.367 | 0.395 | 0.615 | 0.376 | 0.450 | 0.007 | 0.372 |
| MTE | 0.248 | 0.308 | 0.362 | 0.506 | 0.358 | 0.391 | 0.367 | 0.363 |
| DegMat | 0.370 | 0.294 | 0.346 | 0.616 | 0.220 | 0.357 | 0.076 | 0.326 |
| Semantic Entropy | 0.289 | 0.384 | 0.235 | 0.541 | 0.277 | 0.410 | 0.029 | 0.309 |
| EigValLaplacian | 0.346 | 0.264 | 0.296 | 0.600 | 0.152 | 0.283 | 0.075 | 0.288 |

Table 16: LLaMA-8B, PRR (higher is better).

| Method | CoQA | GSM8k | MMLU | Trivia | WMT14FrEn | WMT19DeEn | XSum | mean |
|---|---|---|---|---|---|---|---|---|
| Ours (Exp, FW) | 0.419 | 0.514 | 0.543 | 0.698 | 0.484 | 0.611 | 0.208 | 0.497 |
| Ours (Beta, FW) | 0.423 | 0.506 | 0.543 | 0.698 | 0.486 | 0.615 | 0.210 | 0.497 |
| CoCoA MTE | 0.416 | 0.505 | 0.528 | 0.689 | 0.443 | 0.568 | 0.208 | 0.479 |
| CoCoA MSP | 0.410 | 0.438 | 0.539 | 0.699 | 0.444 | 0.590 | 0.222 | 0.477 |
| CoCoA PPL | 0.421 | 0.471 | 0.539 | 0.683 | 0.442 | 0.573 | 0.209 | 0.477 |
| MSP | 0.338 | 0.386 | 0.548 | 0.680 | 0.328 | 0.420 | 0.177 | 0.411 |
| Consistency | 0.417 | 0.407 | 0.493 | 0.657 | 0.332 | 0.487 | 0.203 | 0.428 |
| MTE | 0.305 | 0.384 | 0.543 | 0.642 | 0.426 | 0.532 | 0.139 | 0.424 |
| SAR | 0.392 | 0.373 | 0.520 | 0.647 | 0.397 | 0.503 | 0.136 | 0.424 |
| PPL | 0.323 | 0.343 | 0.548 | 0.653 | 0.394 | 0.520 | 0.144 | 0.418 |
| DegMat | 0.401 | 0.379 | 0.494 | 0.663 | 0.291 | 0.440 | 0.174 | 0.406 |
| Semantic Entropy | 0.306 | 0.420 | 0.474 | 0.593 | 0.323 | 0.411 | 0.149 | 0.383 |
| EigValLaplacian | 0.379 | 0.348 | 0.471 | 0.656 | 0.241 | 0.400 | 0.175 | 0.381 |

Table 17: Falcon-7B, PRR (higher is better).

