# OpenReview forum: "Multidimensional Uncertainty Quantification via Optimal Transport"
_ICLR.cc/2026/Conference — ICLR 2026 Conference Withdrawn Submission_

### Official Review · Reviewer_D3gc · 2025-10-17

**Soundness:** 3
**Presentation:** 2
**Contribution:** 2
**Rating:** 4
**Confidence:** 4

**Summary:**

The authors present a new ingenious way of combining different uncertainty metrics that involves writing them in a vector form, and then uncertainty is fused using  entropy-regularized optimal transport (OT).

**Strengths:**

The idea is interesting and new, and the experiments seem to support it.

**Weaknesses:**

While very creative, I find the paper to be not ready for publication yet. Naturally this is a personal opinion, and if the majority of the referees are in favor of it, I will abide.

What the paper seems to imply is that if we consider $m$ different positive uncertainty quantification measures, they give rise to an $m$-dimensional vector space, a subset of $\mathbb{R}^m_+$. This is a very profound claim that is not spelled out, nor justified in the paper. While I understand heuristically why this is appealing, still it puzzles me quite a bit: For example, could it not be the case that, since one or more uncertainty measures may be correlated, we need less than $m$ dimensions?

Other things that make me lean towards not accepting it in its current form are the writing, that is too fragmented and could be improved, and the fact that the authors say that $\mu$ is the empirical distribution of the $\mathbf{s}_i$'s. But surely this would give rise to a discrete uniform more often than not, right? This is because two uncertainty measures vectors need to be exactly equal in each component to be counted twice. Wouldn't this be a problem?

Finally, I would like to invite the authors to compare their method to imprecise probabilistic machine learning, as studied by Sale, Hüllermeier, Munadet, Chau, Caprio, Javanmardi, Desterke, Cuzzolin, and more, where the concept of "meta-uncertainty", heuristically similar to the one presented by the authors in the form of a vector of uncertainty measures, is ubiquitous. In particular, even though for different reasons, in that literature OT for uncertainty quantification has been already proposed https://arxiv.org/abs/2410.03267, published in the proceedings of ISIPTA 2025.

Overall, in my opinion this reads like a good paper that needs a few more tweaks and explanations to be ready for publication.

**Questions:**

See Weaknesses

---

> ### Author Response · Authors · 2025-11-18
>
> We would like to thank the reviewer for their feedback and insightful comments.
> Please find below the answers to your comments.
>
> -----
>
> > What the paper seems to imply is that if we consider m different positive uncertainty quantification measures, they give rise to an m-dimensional vector space, a subset of R^m_+ . This is a very profound claim that is not spelled out, nor justified in the paper. While I understand heuristically why this is appealing, still it puzzles me quite a bit: For example, could it not be the case that, since one or more uncertainty measures may be correlated, we need less than m dimensions?
>
> We agree with the reviewer that in the case of significantly correlated predictive uncertainty measures, one indeed needs fewer than m dimensions. However, this does not lead to any problems in terms of ordering, and the method remains robust.
>
> We demonstrated this in an artificial example, in Appendix, Section E, "Extreme Compositions", where we considered several "extreme" cases of choosing the components to fill the vector. We demonstrate that the ordering provided is logical in the sense that for effectively 1D signals, the order is preserved.
>
> In our Tables 1 and 2, where we used risk-based measures, a similar situation happens. Specifically, risk-based measures are highly correlated and have approximately the same distinguishing power. Yet, the aggregation we use allows us to improve upon.
>
> ----
>
> > Other things that make me lean towards not accepting it in its current form are the writing, that is too fragmented and could be improved, and the fact that the authors say that \mu is the empirical distribution of the s_i's.
>
> We thank the reviewer for their feedback on writing. Following the recommendation of other reviewers, we will include a more explicit motivation for considering optimal transport, which will make the transition to the technical details smoother. We are willing to consider incorporating other writing suggestions if the reviewer provides them.
>
> > But surely this would give rise to a discrete uniform more often than not, right? This is because two uncertainty measures vectors need to be exactly equal in each component to be counted twice. Wouldn't this be a problem?
>
> We may not fully understand the concern here. It is, of course, almost surely a discrete uniform measure. That's why we use discrete optimal transport, which essentially considers moving a set of points (following the initial motivation of Monge). That's the essence of our approach, and that is why barycentric projection is needed for new points. If you can elaborate on your concern further, we will be able to provide further clarification.
>
> -----
>
> > Finally, I would like to invite the authors to compare their method to imprecise probabilistic machine learning, as studied by Sale, Hüllermeier, Munadet, Chau, Caprio, Javanmardi, Desterke, Cuzzolin, and more, where the concept of "meta-uncertainty", heuristically similar to the one presented by the authors in the form of a vector of uncertainty measures, is ubiquitous. In particular, even though for different reasons, in that literature OT for uncertainty quantification has been already proposed https://arxiv.org/abs/2410.03267, published in the proceedings of ISIPTA 2025.
>
> We thank the reviewer for sharing the related literature and the specific paper. We are happy to consider other methods in our evaluation, including those the reviewer has sent.
>
> The specific paper the reviewer has shared is exciting, but it is purely theoretical without a publicly available implementation.
> Please correct us if we are wrong, but it appears that the paper addresses a different issue.
> Specifically, the paper extends optimal transport, which is defined between two precise probability distributions, to the case where each side is only partially specified by a lower probability.
>
> However, it does not (at least explicitly) discuss the opportunity of combining different measures of predictive uncertainty into a single scalar. We do not see how one can utilize the theoretical development from the paper for uncertainty quantification in a straightforward manner. Can the reviewer comment on this possibility and explain how it is specifically related to the paper under review?

---

> > ### Comment · Reviewer_D3gc · 2025-11-18
> >
> > Dear Authors,
> >
> > My concerns are solved. What I meant by pointing to https://arxiv.org/abs/2410.03267 is that there exists already a way of transporting sets of probabilities, that are well-studied tools for uncertainty quantification, so referencing to it would strengthen the related literature section.
> >
> > Furthermore, a brief discussion on how uncertainty measures for sets of probabilities such as the generalized Hartley measure, that can be transported using OT techniques, relate to the meta-UQ method that you propose would also strengthen the work (but this is up to you, I will not vote against publication if you do not add it).
> >
> > Finally, I want to mention that OT has also recently been studied in the context of belief functions, another well-studied tool in the UQ literature, see https://www.sciencedirect.com/science/article/pii/S016501142500168X

---

> > > ### Author Response · Authors · 2025-11-19
> > >
> > > Dear Reviewer D3gc,
> > >
> > > Thank you for your prompt reply and for increasing the evaluation score!
> > >
> > > Thank you for clarifying your point. We will add the discussion of the paper you mentioned to the Related Work section in the revision soon and mention relevant works in the field of uncertainty for sets of probabilities in general.
> > >
> > > Thank you,
> > >
> > > Authors

---

### Official Review · Reviewer_LbkW · 2025-10-27

**Soundness:** 4
**Presentation:** 2
**Contribution:** 3
**Rating:** 6
**Confidence:** 4

**Summary:**

This paper introduces VecUQ-QT, an uncertainty quantification (UQ) method that uses multiple UQ metrics to more holistically / robustly evaluate uncertainty. Given a stack of uncertainty measures, VecUQ-QT learns a mapping between an isotropic distribution (either Beta or exponential) and uncertainty vectors on in-distribution (ID) calibration data. Once the mapping is obtained, the scalar uncertainty rank can be determined and used for downstream tasks like out-of-distribution detection.

**Strengths:**

1. Conceptual novelty. Many existing / prior works incorporate various sources of uncertainty (e.g., epistemic, aleatoric) to more holistically capture total uncertainty. However, the general framing / idea of the proposed approach (i.e., Monge-Kantorovich ranks) for learning optimal transport between the ID data and the target distribution is quite novel and interesting.
2. Experimental validation includes a good range of datasets and downstream tasks, which adequately supports the author's theoretical claims.
3. Simplicity. The proposed method is quite simple in its methodology. It only requires a small calibration set.

**Weaknesses:**

1. Theoretical analysis. Authors address the convex-hull limitation by adding outer support anchors using an empirically determined scaling factor. Current analysis / discussion surrounding this is mostly heuristic and would definite for more rigorous theoretical investigation on the trade-offs of this workaround (e.g., are there any formal guarantees on OOD inputs on the extended support?)
2.  Scalability and design of uncertainty vector. The proposed approach counter-balances the weaknesses of certain metrics (e.g., 1-MSP) by leveraging the strengths of other metrics (e.g., Mahalanobis distance). However, there is little discussion regarding strategies for choosing metrics to include. This will presumably have a large impact on the quality of VecUQ-QT. Furthermore, the inclusion of additional metrics will increase the computational load of the overall pipeline---there is a trade-off there that warrants further discussion / investigation.
3. Notation. The equations and notation in Section 2 are a little dense and would greatly benefit from additional context / explanation.

**Questions:**

1. How sensitive is the proposed method to the size of the calibration set? How large does the ID dataset need to be for robust performance?
2. Is the method sensitive or negatively impacted in any way by including highly correlated metrics?

---

> ### Author Response · Authors · 2025-11-18
>
> We would like to thank the reviewer for their feedback and insightful comments.
> Please find below the answers to your comments.
>
> -----
>
>
> > Theoretical analysis. Authors address the convex-hull limitation by adding outer support anchors using an empirically determined scaling factor. Current analysis / discussion surrounding this is mostly heuristic and would definite for more rigorous theoretical investigation on the trade-offs of this workaround (e.g., are there any formal guarantees on OOD inputs on the extended support?)
>
> This is a good comment and a valid remark. Indeed, this idea with outer anchors is heuristic, which allows us to bypass limitations of the barycentric projection. Nevertheless, it does not harm or break any properties of the entropic-regularized optimal transport, as we add these points to the source domain. These points, by construction, have a higher uncertainty component-wise than any other point in the calibration dataset, with a large margin.
>
> In Appendix, Section C, we demonstrate in our ablation study that the choice of margin is not very important. Even for a multiplier of 0 (i.e., no anchors at all), the learned mapping still provides good results, particularly in the OOD detection problem.
>
> ---
>
> > Scalability and design of uncertainty vector. The proposed approach counter-balances the weaknesses of certain metrics (e.g., 1-MSP) by leveraging the strengths of other metrics (e.g., Mahalanobis distance). However, there is little discussion regarding strategies for choosing metrics to include. This will presumably have a large impact on the quality of VecUQ-QT. Furthermore, the inclusion of additional metrics will increase the computational load of the overall pipeline---there is a trade-off there that warrants further discussion / investigation.
>
> Thank you for the great question!
>
> The underlying knowledge about the problem at hand should inform this decision. For example, suppose one knows that the problem in hand is of an epistemic uncertainty nature (e.g., the detection of out-of-distribution inputs). In that case, one should not use measures for detecting aleatoric-high inputs (as we demonstrate in Figure 1), but instead fill in the vector with different components that are individually reasonable for out-of-distribution detection.
>
> If the problem at hand is to quantify uncertainty in the prediction of a specific type of cancer, and it is known that several measures are reasonably effective in this context, one can use them simultaneously. In our experiments, we empirically observe that the ordering of the aggregated uncertainty is never worse than the ordering by the worst component.
>
> If expert knowledge is not available, then the general recommendation is to use measures of a different nature (such as risk-based and density-based) that perform well individually. In this case, the information they provide will complement each other.
>
> Indeed, having more uncertainty measures in a vector requires more computations, but the overhead is typically tiny. All of the uncertainty measures we considered do not require additional forward passes through the model. Therefore, once the prediction vector (or vectors, in the case of an ensemble) is calculated, it can be reused in the computation of uncertainty measures.
>
> As for the algorithm we use to fit entropic-regularized OT, we use the one proposed in Cuturi (2013) [1]. The complexity of the algorithm is O(m^2 n), where m represents the number of stacked uncertainty measures and n represents the size of the calibration dataset.
>
> [1] Cuturi, M. (2013). Sinkhorn distances: Lightspeed computation of optimal transport. Advances in neural information processing systems, 26.
>
> ---
>
> > Notation. The equations and notation in Section 2 are a little dense and would greatly benefit from additional context / explanation.
>
> We agree with the reviewer that Section 2 is technical and dense in notation. However, to better describe the mathematical derivations and write precisely what we do, we have to introduce all this notation.
>
> Adding plain word context would indeed ease reading, and we will make corresponding changes to the text soon. Thank you for the suggestion!

---

> ### Author Response · Authors · 2025-11-18
>
> Questions:
>
> > How sensitive is the proposed method to the size of the calibration set? How large does the ID dataset need to be for robust performance?
>
> We did not conduct this experiment, and this is indeed a great direction to investigate. We will include an experiment that demonstrates the connection between calibration dataset size and the performance on a downstream problem as soon as we have a result.
>
>
> > Is the method sensitive or negatively impacted in any way by including highly correlated metrics?
>
> This is a great question!
>
> In fact, we conducted the corresponding experiment and found no negative impact on the proposed method in cases where the measures in the vector are highly correlated or even coincident. Specifically in Section E, "Extreme Compositions", we considered several "extreme" cases of choosing the components to fill the vector in. We demonstrate that the ordering provided is logical in the sense that for effectively 1D signals, the order is preserved, as should be in the case of vanilla OT ordering.

---

> ### Author Response · Authors · 2025-11-27
>
> Dear Reviewer LbkW,
>
> Following the Reviewers' great question, we conducted an additional experiment to investigate the sensitivity of our approach to the size of calibration data. We retain the same individual components as in the main text of the paper (in Tables 1 and 2), and change the size of the calibration data used for the fitting of optimal transport (in an unsupervised manner).
>
>
> For all datasets, the calibration datasets were fixed at 1800 objects (100%). This column (100%) aligns with the results presented in the main text in Tables 1 and 2. Then, we change the size of the calibration dataset.
>
> We observe that even for 10% (only 180 objects from the split of the calibration dataset), the results remain competitive with the original one. For extremely small calibration datasets (<1%, or equivalently, <18 objects from the data split), the results start to deteriorate, especially for datasets other than CIFAR-10.
>
> We believe that this demonstrates the robustness of our approach to the size of the calibration dataset.
>
>
> | ind_dataset| eval| 100%| 70% | 50%| 20%| 10%| 1%| 0.5%|
> |-----|------|-----|----------------|----------------|----------------|----------------|----------------|----------------|
> | cifar10| cifar10 [miscls]| 0.944 ± 0.002  | 0.944 ± 0.002  | 0.944 ± 0.002  | 0.944 ± 0.002  | 0.944 ± 0.002  | 0.944 ± 0.002  | 0.941 ± 0.002  |
> | cifar10| cifar10 [selective]| 0.997 ± 0.000  | 0.997 ± 0.000  | 0.997 ± 0.000  | 0.997 ± 0.000  | 0.997 ± 0.000  | 0.997 ± 0.000  | 0.997 ± 0.000  |
> | cifar10| cifar100 [ood]| 0.918 ± 0.001  | 0.918 ± 0.001  | 0.918 ± 0.001  | 0.918 ± 0.001  | 0.918 ± 0.001  | 0.917 ± 0.002  | 0.916 ± 0.003  |
> | cifar10| svhn [ood]| 0.957 ± 0.005  | 0.957 ± 0.005  | 0.957 ± 0.005  | 0.958 ± 0.005  | 0.958 ± 0.005  | 0.956 ± 0.006  | 0.952 ± 0.006  |
> | cifar10| tiny_imagenet [ood]| 0.912 ± 0.001  | 0.912 ± 0.000  | 0.912 ± 0.000  | 0.913 ± 0.001  | 0.912 ± 0.001  | 0.911 ± 0.002  | 0.910 ± 0.004  |
> | cifar100| cifar10 [ood]| 0.765 ± 0.001  | 0.764 ± 0.001  | 0.763 ± 0.001  | 0.764 ± 0.002  | 0.760 ± 0.003  | 0.762 ± 0.003  | 0.634 ± 0.021  |
> | cifar100| cifar100 [miscls]| 0.849 ± 0.003  | 0.849 ± 0.002  | 0.848 ± 0.003  | 0.848 ± 0.004  | 0.846 ± 0.004  | 0.848 ± 0.002  | 0.703 ± 0.031  |
> | cifar100| cifar100 [selective]| 0.916 ± 0.001  | 0.915 ± 0.001  | 0.915 ± 0.001  | 0.915 ± 0.001  | 0.914 ± 0.001  | 0.916 ± 0.001  | 0.867 ± 0.010  |
> | cifar100| svhn [ood]| 0.870 ± 0.006  | 0.870 ± 0.006  | 0.868 ± 0.007  | 0.870 ± 0.007  | 0.864 ± 0.006  | 0.854 ± 0.006  | 0.738 ± 0.018  |
> | cifar100| tiny_imagenet [ood]| 1.000 ± 0.000  | 1.000 ± 0.000  | 1.000 ± 0.000  | 1.000 ± 0.000  | 1.000 ± 0.000  | 1.000 ± 0.000  | 0.995 ± 0.006  |
> | tiny_imagenet  | tiny_imagenet [miscls]| 0.847 ± 0.002  | 0.843 ± 0.002  | 0.841 ± 0.003  | 0.843 ± 0.003  | 0.839 ± 0.003  | 0.842 ± 0.004  | 0.720 ± 0.018  |
> | tiny_imagenet  | tiny_imagenet [selective] | 0.886 ± 0.001  | 0.884 ± 0.001  | 0.883 ± 0.001  | 0.884 ± 0.001  | 0.882 ± 0.002  | 0.885 ± 0.001  | 0.835 ± 0.007  |
> | tiny_imagenet  | imagenet_a [ood]| 0.847 ± 0.002  | 0.846 ± 0.002  | 0.845 ± 0.002  | 0.846 ± 0.002  | 0.843 ± 0.002  | 0.840 ± 0.005  | 0.744 ± 0.014  |
> | tiny_imagenet  | imagenet_r [ood]| 0.835 ± 0.001  | 0.832 ± 0.001  | 0.831 ± 0.001  | 0.832 ± 0.001  | 0.830 ± 0.001  | 0.828 ± 0.006  | 0.727 ± 0.016  |
> | tiny_imagenet  | imagenet_o [ood]| 0.760 ± 0.002  | 0.762 ± 0.001  | 0.763 ± 0.002  | 0.762 ± 0.002  | 0.763 ± 0.002  | 0.764 ± 0.003  | 0.712 ± 0.010  |
>
> ----
> ----
>
>
> ----
> ----
>
> In light of this additional experiment and the Reviewer's earlier positive comments, we would also like to ask whether the Reviewer is willing to revisit their overall score.
>
> The Reviewer was initially the most positive about the paper, and Reviewer D3gc has since reconsidered and increased their rating following the rebuttal.
>
> We want to emphasize that all Reviewers currently recognize the novelty and creativity of the approach proposed in the paper. We do believe that this paper is a step in an important direction. Meaningful enough to be shared with the community.
>
> If, after seeing the new robustness results, the Reviewer feels that the contribution is closer to an Accept than to a Weak Accept, we would be very grateful if they could reflect this in their overall assessment.
>
> A stronger recommendation from the Reviewer would help the AC form a balanced view of the submission, given the differing levels of detail and engagement across reviews.
>
> Of course, we fully understand if the Reviewer prefers to keep their current score.
>
> In any case, we sincerely appreciate the time and care the Reviewer has devoted to reading our work, asking insightful questions, and helping us improve the paper.
>
> We are also very happy to answer any further questions the Reviewer may have.
>
> Best regards,
>
> Authors

---

### Official Review · Reviewer_7cU2 · 2025-10-29

**Soundness:** 1
**Presentation:** 1
**Contribution:** 2
**Rating:** 2
**Confidence:** 4

**Summary:**

The paper proposes stacking multiple uncertainty measures into a vector and ordering these vectors via entropy-regularized optimal transport (OT) to a simple reference distribution. The final scalar is the distance from the reference center. The method is evaluated across several tasks, including misclassification detection, OOD detection, selective prediction, and selective generation, and often stays strong even when one component is weak.

**Strengths:**

+ The idea of a multidimensional view on uncertainty quantification is relatively new and interesting.
+ The empirical results are consistently competitive across various tasks.

**Weaknesses:**

- The paper lacks a clear motivation and solid theoretical analysis. It is unclear why OT is considered and why the radial order to a reference can be used as an uncertainty score. No guarantee that the radial order preserves or improves any proper scoring rule. And there is no principled criterion for the choice of reference distribution.
- The paper claims 'our multidimensional framework supports non-additive aggregation of AU and EU', but the experiments did not explicitly support this claim. It is unclear how AU and EU are operationally measured, and non-additively aggregated here.
- The computational cost of stacking multiple uncertainty measures and the ranking algorithm is hand-waved.

**Questions:**

See weaknesses.

---

> ### Author Response · Authors · 2025-11-18
>
> We would like to thank the reviewer for their feedback and insightful comments.
> Please find below the answers to your comments.
>
> -----
>
>
> > idea of a multidimensional view on uncertainty quantification is relatively new
>
> We are pleased to hear the reviewer's evaluation that recognizes the novelty of our proposed approach. We would like to kindly point out that, to the best of our knowledge, our approach is novel.
>
> Specifically, we are not aware of any other papers that explicitly target aggregating several uncertainty measures of arbitrary nature to account for all the signals they convey. We think this is a crucial point to emphasize.

---

> ### Author Response · Authors · 2025-11-18
>
> > **The paper lacks a clear motivation and solid theoretical analysis. It is unclear why OT is considered and why the radial order to a reference can be used as an uncertainty score. No guarantee that the radial order preserves or improves any proper scoring rule. And there is no principled criterion for the choice of reference distribution.**
>
> -----
>
> > “lacks a clear motivation and solid theoretical analysis. It is unclear why OT is considered”
>
> We kindly disagree with this assessment and hope the following comment clarifies our motivation and the particular design choice.
>
> **Motivation:**
>
> We aim to aggregate several uncertainty measures into a single measure. Here is one of the reasons why we want to do it. Several recent studies [1, 2, 3] introduce a framework based on the risk interpretation of uncertainty, which allows for the generation of a large number (~80) of uncertainty measures (and many more exist). These works studied these measures extensively in different downstream problems. One of the conclusions is that each downstream problem (and even a dataset for a particular problem) requires a specific measure of uncertainty. However, it is not clear in advance how to choose a particular one, as different instantiations of proper scoring rules capture different aspects, and various choices of Bayesian approximations also capture the interaction between predictions differently. Given the plethora of these measures, the practitioner must choose a single one, putting all "eggs in one basket". With the methodology we propose, the practitioner is not forced to choose a single uncertainty measure, but can select a set of reasonable ones. The specific approach to aggregating these measures, as demonstrated in our experiments, leads to more robust performance.
>
> **Theoretical analysis and choice of OT:**
>
> The specific choice of OT and the specific radial order are not heuristics we introduce in this paper. This choice is determined by recent advancements in the area of developing order over vectors [see 4, 5, 6], especially in the context of multivariate quantile regression.
> We cite these works in line 118. We follow their ordering methodology in our paper.
>
> The idea of using OT is to induce an order over vectors. The ordering of the scalars is a trivial problem (as scalars have a natural notion of ordering, with the < and > relation). However, it is way more ambiguous with vectors. There is no standard agreement on the "correct" ordering of vectors. However, the studies we cited provide one way to induce it.
>
> Several alternatives to the OT-based ordering exist [4]. But the ordering provided by the solution of optimal transport has more advantages than others. Moreover, it is non-parametric, and no uncertainty is associated with the parameters.
>
> We agree with the reviewer that this discussion is essential for a complete understanding of the picture. And we will add it to the main text. However, it is thoroughly covered in the cited works mentioned in line 118. These papers also discuss alternative definitions of vector ordering and explain why the OT-based approach is better.
>
>
> [1] Kotelevskii, N., Kondratyev, V., Takáč, M., Moulines, E., & Panov, M. From Risk to Uncertainty: Generating Predictive Uncertainty Measures via Bayesian Estimation. In The Thirteenth International Conference on Learning Representations.
>
> [2] Schweighofer, K., Aichberger, L., Ielanskyi, M., & Hochreiter, S. On Information-Theoretic Measures of Predictive Uncertainty. In The 41st Conference on Uncertainty in Artificial Intelligence.
>
> [3] Hofman, P., Sale, Y., & Hüllermeier, E. (2024). Quantifying aleatoric and epistemic uncertainty with proper scoring rules. arXiv preprint arXiv:2404.12215.
>
> [4] Marc Hallin and Dimitri Konen. Multivariate quantiles: Geometric and measure-transportation-based contours. In Applications of Optimal Transport to Economics and Related Topics, pp. 61–78. Springer, 2024.

---

> ### Author Response · Authors · 2025-11-18
>
> > “why the radial order to a reference can be used as an uncertainty score”
>
> In our paper, we follow the approach proposed in [4, 5, 6], which suggests optimal transport-based ordering over vectors.
>
> Generally, it is of interest to construct statistics that have a certain ground-truth meaning, like, for example, the expected error of prediction. However, in practice, multiple different estimates of such uncertainty measures are noisy and often provide complementary signals. Thus, this paper, instead of trying to approximate a single ground truth uncertainty measure, aims to fuse several complementary signals about the uncertainty of prediction with the aim of achieving the best possible ordering of test data.
>
> One of the good properties we require in general from uncertainty quantification measures (of any type) is the ordering they produce. In scientific papers, the evaluation of these measures is conducted, taking the order into account, rather than focusing on the exact absolute values. For example, when evaluating out-of-distribution detection as a downstream problem, we typically compare the order induced by a measure under consideration. We evaluate it on in-distribution objects and out-of-distribution objects, and then compute the ROC-AUC (which is value-agnostic and takes only order into account). The same logic applies to misclassification detection, selective prediction, and active learning.
>
> In other applications, where a decision needs to be made based on the uncertainty score, the order is again the only factor that matters. In these cases, the typical workflow is to select a fixed threshold for the uncertainty value, and when it is exceeded, abstain from making a prediction (which is too unreliable due to high uncertainty) and then take some action (e.g., refer it to an expert).
>
> The values we have as a result of our approach are "soft rank", which we assign to initial vectors of uncertainty scores to order them. There might not be much "physical" sense in the values themselves. However, as our empirical evaluation reveals, the order these values induce is reasonable and yields robust results across various datasets/problems. This is the reason why we can consider the radial order as an uncertainty measure.
>
> > No guarantee that the radial order preserves or improves any proper scoring rule.
>
> In fact, we have the guarantee of order preservation.
>
> To better explain it, we need to make a brief remark on the solution provided by optimal transport. In classical discrete optimal transport with a quadratic cost function, the optimal map between two distributions is known in its form. It can be represented as the gradient of a convex function (a classical result). For the one-dimensional case, it means that the map will be monotone. Therefore, the ordering before and after the map is preserved. One may think of it as a shift.
>
> For the vector-valued functions, there is no natural definition of “monotone function”, but there is a generalization of this term, namely “cyclical monotonicity”. The optimal solution to classical discrete optimal transport with a quadratic cost function has this property.
>
> However, classical optimal transport does not provide a method for generalizing to new, unseen points, which is crucial for practical considerations. Therefore, it becomes impractical. However, entropic regularized optimal transport allows us to make mappings for new, unseen inputs. The solution of entropic regularized optimal transport coincides with the classical optimal transport in the case when epsilon equals 0. Therefore, it approximately preserves cyclical monotonicity for small values of epsilon and enjoys all the properties of plain OT.
>
> As a specific instantiation of the optimal transport algorithm, we considered entropic-regularized OT. The reason for this is that, unlike the standard formulation, it allows for generalization to new, unseen points.
>
> To investigate the impact of entropic-regularized optimal transport on the ordering provided by the learned mapping, we conducted a detailed ablation study. Specifically in Section E, "Extreme Compositions", we considered several "extreme" cases of choosing the components to fill the vector in. We demonstrate that the ordering provided is logical in the sense that for effectively 1D signals, the order is preserved, as should be in the case of vanilla OT ordering.
>
> We hope this addresses the reviewer's concern. We are happy to elaborate on further questions.

---

> ### Author Response · Authors · 2025-11-18
>
> > “And there is no principled criterion for the choice of reference distribution.”
>
> We agree with the reviewer that we did not provide a criterion for selecting the reference distribution. In our paper, we empirically tested two simple and computationally inexpensive distributions, demonstrating that they yield good results. In the original papers [4, 5, 6], arbitrary sign components were considered, and therefore the target distribution was chosen to be a ball. In our case, since most of the uncertainty measures in classification are positive, we considered distributions defined over a positive domain. As our results show, a choice of Beta distribution typically yields robust and good results, and can be a safe default choice.
>
> References:
>
> [1] Kotelevskii, N., Kondratyev, V., Takáč, M., Moulines, E., & Panov, M. From Risk to Uncertainty: Generating Predictive Uncertainty Measures via Bayesian Estimation. In The Thirteenth International Conference on Learning Representations.
>
> [2] Schweighofer, K., Aichberger, L., Ielanskyi, M., & Hochreiter, S. On Information-Theoretic Measures of Predictive Uncertainty. In The 41st Conference on Uncertainty in Artificial Intelligence.
>
> [3] Hofman, P., Sale, Y., & Hüllermeier, E. (2024). Quantifying aleatoric and epistemic uncertainty with proper scoring rules. arXiv preprint arXiv:2404.12215.
>
> [4] Marc Hallin and Dimitri Konen. Multivariate quantiles: Geometric and measure-transportation-based contours. In Applications of Optimal Transport to Economics and Related Topics, pp. 61–78. Springer, 2024.
>
> [5] Marc Hallin, Eustasio del Barrio, Juan Cuesta-Albertos, and Carlos Matran. Distribution and quantile functions, ranks and signs in dimension d: A measure transportation approach. Annals of Statistics, 49(2): 1139–1165, 2021.
>
> [6] Victor Chernozhukov, Alfred Galichon, Marc Hallin, and Marc Henry. Monge-kantorovich depth, quantiles, ranks, and signs. Annals of Statistics, 45(1):223–256, 2017.

---

> ### Author Response · Authors · 2025-11-18
>
> > The paper claims 'our multidimensional framework supports non-additive aggregation of AU and EU', but the experiments did not explicitly support this claim. It is unclear how AU and EU are operationally measured, and non-additively aggregated here.
>
>
> We have conducted an experiment with the additive measure of components for the out-of-distribution detection problem, and we are ready to consider other problems / measure combinations if required. Here, specifically, we chose conjugate estimates or AU and EU (R_b^1 and R_exc^1,1) that both sum to the TU (R_tot^(1,1)).
>
> The results we have:
>
> | In-dist dataset | OOD dataset | Method   | Score (↑)            |
> |-----------------|------------|----------|----------------------|
> | CIFAR-10        | CIFAR-100  | VecUQ-OT | **0.9138 ± 0.0006**  |
> | CIFAR-10        | CIFAR-100  | TU (sum) | 0.9115 ± 0.0005      |
> | TinyImageNet    | ImageNet-O | VecUQ-OT | **0.7565 ± 0.0031**  |
> | TinyImageNet    | ImageNet-O | TU (sum) | 0.7535 ± 0.0017      |
> | TinyImageNet    | ImageNet-R | VecUQ-OT | 0.8360 ± 0.0011      |
> | TinyImageNet    | ImageNet-R | TU (sum) | **0.8366 ± 0.0015**  |
>
>
>
> From the results, we see that the ordering indeed changes for all of the datasets. While the performance for these two combinations is close, our method provides a slight improvement on some datasets. Below, we explain why the order changed.
>
> Since our approach allows for the non-additive aggregation of AU and EU, it will not be equivalent to simply adding them together. From the construction of the final uncertainty measure, the resulting ordering will be determined as a radial distance from the origin to the image of the transport map, which is a nonlinear aggregation.
>
> The additive aggregation of uncertainty, although theoretically correct, has been recently criticized [7] (see Section 4.4 therein) for not holding in practical machine learning scenarios. Our approach is one of the possible methods for achieving non-additive aggregation.
>
> The estimates of AU and EU can be obtained in an arbitrary manner, and in principle, they can even be of different natures (AU from an information-theoretic perspective, EU from a density-based perspective). In this case, when AU and EU estimates have different natures and scales, in the case of additive aggregation, one needs to tune the coefficients of aggregation. In our case, everything is fitted in an unsupervised manner.
>
> [7] Wimmer, L., Sale, Y., Hofman, P., Bischl, B., & Hüllermeier, E. (2023, July). Quantifying aleatoric and epistemic uncertainty in machine learning: Are conditional entropy and mutual information appropriate measures?. In Uncertainty in artificial intelligence (pp. 2282-2292). PMLR.
>
> -----
>
> > The computational cost of stacking multiple uncertainty measures and the ranking algorithm is hand-waved.
>
> As we write in line 262, the entropic regularization yields scalable solvers via Sinkhorn iterations. Specifically, the algorithm used to solve the problem is the Sinkhorn-Knopp matrix scaling algorithm, as proposed in Cuturi (2013) [8]. The complexity of the algorithm is O(m^2 n), where m represents the number of stacked uncertainty measures and n represents the size of the calibration dataset.
>
> [8] Cuturi, M. (2013). Sinkhorn distances: Lightspeed computation of optimal transport. Advances in neural information processing systems, 26.

---

> ### Author Response · Authors · 2025-11-24
>
> Dear Reviewer 7cU2,
>
> We would like to thank you once again for the valuable feedback you provided!
>
> We understand that the reviewer's duty overlaps with their own ICLR submission rebuttal, and possibly other conferences.
>
> However, since only a bit more than one week is left until the end of the authors-reviews discussion, it is essential to hear your feedback on the authors' response.
>
> We believe that the most crucial concerns, specifically the choice of OT and its justification, as well as the addition of an additive baseline, were addressed in our response. We also described which specific algorithm we used and indicated its complexity.
> We are more than happy to provide answers to any further questions the reviewer might have.
>
> Best regards,
>
> Authors

---

> ### Author Response · Authors · 2025-11-28
>
> Dear Reviewer 7cU2,
>
> There is less than a week left till the end of the Authors-Reviewers discussion period. However, we have not received any comments from you till the moment.
>
> We posted our replies to your review **10 days ago**, but have not received your answer.
>
> We have addressed all the concerns you have raised, and we are willing to elaborate on any further questions you may have.
>
> Please see our answers and let us know if you still have any concerns.
>
> To summarize:
> 1. We explained the motivation behind OT-based ordering.
> 2. We conducted two additional experiments that demonstrated that our approach allows for aggregating uncertainties in a non-additive manner. See our previous response to your comments. Additionally, for a more specific request from the Reviewer qyiv, we aggregated measures in Tables 1 and 2 in an additive manner, and our method still demonstrates robustness.
> 3. We specified the complexity of the algorithm.
>
> We are looking forward to your response.
>
> Best regards,
>
> Authors

---

### Official Review · Reviewer_qyiv · 2025-10-31

**Soundness:** 1
**Presentation:** 2
**Contribution:** 3
**Rating:** 2
**Confidence:** 4

**Summary:**

This paper proposes a new method for uncertainty quantification using vectors of uncertainty values. These vectors can be constructed by computing the uncertainty of an input using different uncertainty measures. The resulting vector is then used to generate uncertainty values by using a rank map fit using entropic optimal transport with calibration data. Practically, this procedure is expanded by additional “outer anchor” points to enforce OOD inputs, which should have uncertainty scores outside of the calibration range, to map to targets points on the boundary. The approach is evaluated in synthetic, image, and text settings across multiple uncertainty tasks in both a qualitative and quantitive manner.

**Strengths:**

- The paper covers the relevant topic of uncertainty quantification.
- The proposed approach is novel and raises interesting discussion.
- The paper features a wide range of experiments.

**Weaknesses:**

- The paper mentions that validation data is split into calibration data and test data, with the calibration data being used to compute the transport map. However, the models used for the baseline uncertainty measures do not get access to this calibration data. This means that the comparison is not fair.
- The proposed method should be evaluated against the (simple) baseline of summing the respective uncertainty measures that are used to construct the vectors.
- There should be a discussion of what it means to rank inputs based on such a vector of uncertainty values.

**Questions:**

- How do you select which uncertainty measures to use to construct the vector? It would be good to add some guidelines about when you should choose which uncertainty measures.
- Let’s say one uses vectors of AU, and EU based on the log loss / entropy, which according to an additive decomposition would sum to TU. Would the ranking of the [AU, EU] vector induced by your approach be the same as the ranking induced by using just TU?
- Is there an (intuitive) meaning behind the uncertainty values that are obtained from a vector of uncertainty values of different uncertainty measures?

---

> ### Author Response · Authors · 2025-11-18
>
> We would like to thank the reviewer for their feedback and insightful comments.
> Please find below the answers to your comments.
>
> ----
>
> > The paper mentions that validation data is split into calibration data and test data, with the calibration data being used to compute the transport map. However, the models used for the baseline uncertainty measures do not get access to this calibration data. This means that the comparison is not fair.
>
>
> As the reviewer mentioned “models” in the comment, we first would like to clarify that we utilize the same predictive model (ensemble) to calculate uncertainties. The moment when calibration data (used for optimal transport) pops up is at the stage of fitting the plan. The model itself is not changed.
>
> Second, it is essential to emphasize that the fitting of the optimal transport map is an unsupervised problem (as stated in line 277) and does not require any supervised labels.
>
> The reviewer is correct that optimal transport needs calibration data to fit the plan. However, we believe this does not make the comparison unfair.
>
> Here is the explanation of why this is so:
>
> 1. All the risk-based measures we use in our comparison do not require any fitting and therefore cannot utilize additional data. They take predicted probability vectors as input and provide ready-to-use output. Consequently, it is unclear how one can use additional data.
> 2. The density-based measures indeed utilize additional data (which requires labels and differs from the calibration data). Still, they have this data in both cases: when we use optimal transport aggregation and when we use the density measure as is, as a 1D measure. This additional dataset is approximately the same size as the one we use for calibration.
>
> We are ready to consider the experimental setup, where the density-based measures are given the calibration data (with labels). However, this will make the comparison unfair in favor of density-based measures. Even in this case, we do not expect the results to change in essence.
>
> That being said, the scenario we use is fair and reflects a practical setup. Indeed, to fit the optimal plan, one needs additional (in-distribution, unlabeled) data. No explicit out-of-distribution data or additional labeled data is provided, and all the algorithms, therefore, are in the same conditions.
>
> We do not have a good idea how one can use in-distribution “calibration data” to improve the results of 1-d scores, especially those that are not fittable (e.g., information-theoretic or risk-based ones).
>
> -----
>
>
> > The proposed method should be evaluated against the (simple) baseline of summing the respective uncertainty measures that are used to construct the vectors.
>
> > Q: Let’s say one uses vectors of AU, and EU based on the log loss / entropy, which according to an additive decomposition would sum to TU. Would the ranking of the [AU, EU] vector induced by your approach be the same as the ranking induced by using just TU?
>
>
> We have conducted an experiment with the additive measure of components for the out-of-distribution detection problem, and we are ready to consider other problems / measure combinations if required. As components, we used AU and EU, as suggested by the reviewer. TU is their sum.
>
> The results we have:
>
> | In-dist dataset | OOD dataset | Method   | Score (↑)            |
> |-----------------|------------|----------|----------------------|
> | CIFAR-10        | CIFAR-100  | VecUQ-OT | **0.9138 ± 0.0006**  |
> | CIFAR-10        | CIFAR-100  | TU (sum) | 0.9115 ± 0.0005      |
> | TinyImageNet    | ImageNet-O | VecUQ-OT | **0.7565 ± 0.0031**  |
> | TinyImageNet    | ImageNet-O | TU (sum) | 0.7535 ± 0.0017      |
> | TinyImageNet    | ImageNet-R | VecUQ-OT | 0.8360 ± 0.0011      |
> | TinyImageNet    | ImageNet-R | TU (sum) | **0.8366 ± 0.0015**  |
>
>
>
> From the results, we see that the ordering indeed changes for all of the datasets. While the performance for these two combinations is close, our method provides a slight improvement on some datasets. Below, we explain why the order changed.
>
> Since our approach allows for the non-additive aggregation of AU and EU, it will not be equivalent to simply adding them together. From the construction of the final uncertainty measure, the resulting ordering will be determined as a radial distance from the origin to the image of the transport map, which is a nonlinear aggregation.
> The additive aggregation of uncertainty, although theoretically correct, has been recently criticized [1] (see Section 4.4 therein) for not holding in practical machine learning scenarios. Our approach is one of the possible methods for achieving non-additive aggregation.
>
> [1] Wimmer, L., Sale, Y., Hofman, P., Bischl, B., & Hüllermeier, E. (2023, July). Quantifying aleatoric and epistemic uncertainty in machine learning: Are conditional entropy and mutual information appropriate measures?. In Uncertainty in artificial intelligence (pp. 2282-2292). PMLR.

---

> ### Author Response · Authors · 2025-11-18
>
> > There should be a discussion of what it means to rank inputs based on such a vector of uncertainty values.
>
> > Q: Is there an (intuitive) meaning behind the uncertainty values that are obtained from a vector of uncertainty values of different uncertainty measures?
>
> Generally, it is of interest to construct statistics that have a certain ground-truth meaning, like, for example, the expected error of prediction. However, in practice, multiple different estimates of such uncertainty measures are noisy and often provide complementary signals. Thus, this paper, instead of trying to approximate a single ground truth uncertainty measure, aims to fuse several complementary signals about the uncertainty of prediction with the aim of achieving the best possible ordering of test data.
>
> One of the good properties we require in general from uncertainty quantification measures (of any type) is the ordering they produce. In scientific papers, the evaluation of these measures is conducted, taking the order into account, rather than focusing on the exact absolute values. For example, when evaluating out-of-distribution detection as a downstream problem, we typically compare the order induced by a measure under consideration. We evaluate it on in-distribution objects and out-of-distribution objects, and then compute the ROC-AUC (which is value-agnostic and takes only order into account). The same logic applies to misclassification detection, selective prediction, and active learning.
>
> In other applications, where a decision needs to be made based on the uncertainty score, the order is again the only factor that matters. In these cases, the typical workflow is to select a fixed threshold for the uncertainty value, and when it is exceeded, abstain from making a prediction (which is too unreliable due to high uncertainty) and then take some action (e.g., refer it to an expert).
>
> The values we have as a result of our approach are "soft rank" (more on this below), which we assign to initial vectors of uncertainty scores to order them. There might not be much "physical" sense in the values themselves. However, as our empirical evaluation reveals, the order these values induce is reasonable and yields robust results across various datasets/problems.

---

> ### Author Response · Authors · 2025-11-18
>
> ===== Why do we consider Optimal Transport? =====
>
> First, we need to make a brief remark on the solution provided by optimal transport. In classical optimal transport with a quadratic cost function, the optimal map between two distributions is known in its form. It can be represented as the gradient of a convex function. For the one-dimensional case, it means that the map will be monotone. Therefore, the ordering before and after the map is preserved.
> For the vector-valued functions, there is no natural definition of "monotone function", but there is a generalization of this term, namely "cyclical monotonicity". The optimal solution to classical discrete optimal transport with a quadratic cost function has this property.
>
> However, classical optimal transport does not provide a method for generalizing to new, unseen points, which is crucial for practical considerations. Therefore, it becomes impractical. However, entropic regularized optimal transport allows us to make mappings for new, unseen inputs. The solution of entropic regularized optimal transport coincides with the classical optimal transport in the case when epsilon equals 0. Therefore, it approximately preserves cyclical monotonicity for small values of epsilon and enjoys all the properties of plain OT.
>
> To investigate the impact of entropic regularized optimal transport on the ordering provided by the learned mapping, we conducted a detailed ablation study. Specifically in Section E, "Extreme Compositions", we considered several "extreme" cases of choosing the components to fill the vector in. We demonstrate that the ordering provided is logical in the sense that for effectively 1D signals, the order is preserved, as should be in the case of vanilla OT ordering.
>
> We would like to reiterate that this particular method of ordering is not the heuristic we propose in the paper. There is a line of theoretical works [2, 3, 4] (cited in our paper) that develops a notion of ordering over vectors. Several alternatives exist [2], but the ordering provided by the solution of optimal transport has more advantages than others. Moreover, it is nonparametric, and no uncertainty is associated with the parameters.
>
> References:
>
> [2] Marc Hallin and Dimitri Konen. Multivariate quantiles: Geometric and measure-transportation-based contours. In Applications of Optimal Transport to Economics and Related Topics, pp. 61–78. Springer, 2024.
>
> [3] Marc Hallin, Eustasio del Barrio, Juan Cuesta-Albertos, and Carlos Matran. Distribution and quantile functions, ranks and signs in dimension d: A measure transportation approach. Annals of Statistics, 49(2): 1139–1165, 2021.
>
> [4] Victor Chernozhukov, Alfred Galichon, Marc Hallin, and Marc Henry. Monge-Kantorovich depth, quantiles, ranks, and signs. Annals of Statistics, 45(1):223–256, 2017.

---

> ### Author Response · Authors · 2025-11-18
>
> > How do you select which uncertainty measures to use to construct the vector? It would be good to add some guidelines about when you should choose which uncertainty measures.
>
> Thank you for the great question!
>
> The underlying knowledge about the problem at hand should inform this decision. For example, suppose one knows that the problem in hand is of an epistemic uncertainty nature (e.g., the detection of out-of-distribution inputs). In that case, one should not use measures for detecting aleatoric-high inputs (as we demonstrate in Figure 1), but instead fill in the vector with different components that are individually reasonable for out-of-distribution detection.
> If the problem at hand is to quantify uncertainty in the prediction of a specific type of cancer, and it is known that several measures are reasonably effective in this context, one can use them simultaneously. In our experiments, we empirically observe that the ordering of the aggregated uncertainty is never worse than the ordering by the worst component.
>
> If expert knowledge is not available, then the general recommendation is to use measures of a different nature (such as risk-based and density-based) that perform well individually. In this case, the information they provide will complement each other.

---

> ### Author Response · Authors · 2025-11-24
>
> Dear Reviewer qyiv,
>
> We would like to thank you once again for the valuable feedback you provided!
>
> We understand that the reviewer's duty overlaps with their own ICLR submission rebuttal, and possibly other conferences.
>
> However, since only a bit more than one week is left until the end of the authors-reviews discussion, it is essential to hear your feedback on the authors' response.
>
> We believe that the most crucial concerns, specifically the choice of OT and its justification, as well as the addition of an additive baseline, were addressed in our response.
> We are more than happy to provide answers to any further questions the reviewer might have.
>
>
> Best regards,
>
> Authors

---

> ### Comment · Reviewer_qyiv · 2025-11-25
>
> I thank the authors for their detailed response and will respond to the main points below.
>
> ### Fairness of comparison
> I will focus here on the comparison of the risk-based measures versus your proposed measures.
> I understand that the same predictive model is used for risk-based measures and the VecUQ-OT approach, and that the calibration step does not require labels. However, this does not fully resolve my concerns about fairness. The problem is that the aforementioned approaches have a different “data-budget”. A fair comparisons in terms of data would be the following. Split the training data into proper training data and calibration data, and train model A on the original training data and model B only on the proper training data. Then, use the risk-based measures with model A and use VecUQ-OT with model B and fit it on the calibration data.
> I would argue that the current comparison is best viewed as an ablation that isolates the effect of the uncertainty measures, while keeping the predictive model fixed, which is still valuable.
>
> Having said that, I understand that in settings which are becoming more relevant these days (e.g. LLMs), it may not be possible to re-train the model, while calibration data may be freely available. The experiments are therefore valuable on their own. However, in settings where one can re-train, which, in the current work, seems to be everything except the text datasets, experiments with a fair comparisons in terms of data should be provided. Without such an evaluation, I find the claim that VecUT-OT performs better than others too much.
>
> ### Comparison to additive baseline
> The comparison I proposed (in the weaknesses) involves simply summing the components that are used with VecUQ-OT. For example, in Sec. 5.2, out-of-distribution detection, this would mean using the sum of total, excess, Bayes risk, and Mahalanobis score. The reason I asked for this, is that summation is a simple way of aggregating uncertainty signals and evaluating against such a simple approach would help clarify the added benefit of using the more sophisticated approach presented here.
>
> Nevertheless, I appreciate the provided results, which give an (empirical) answer to the question I raised regarding additivity of the AU and EU components of the measures based on entropy.
>
> ### Selection of uncertainty measures
> It is clear that the approach, which allows for combining different uncertainty measures, will take the different characteristics of the output of the predictive model into account. However, with the merging of this information on the basis of optimal transport, it is difficult to say when these individual signals will complement each other. This was also the reason for recommending the baseline of summing the different uncertainty signals, which at least gives a more interpretable notion of uncertainty that may be better understood. Though, the empirical evidence suggests that using the optimal transport approach can be beneficial.
>
>
> I would like to emphasize that the direction of this work is very interesting, but, in its current state, the work is not ready for acceptance.

---

> > ### Author Response · Authors · 2025-11-25
> >
> > Dear Reviewer qyiv,
> >
> > Thank you for your response.
> >
> > Before we begin the suggested experiment and address other points in the Reviewer's message, we have a quick question to clarify our understanding of the experimental setup the Reviewer implies.
> >
> > Is it true that [Original training data] == [proper training data] + [calibration data]?
> >
> > And that Reviewer suggests to train model A with [Original training data] (therefore with **labeled** [calibration data]), and model B only with [proper training data], and calibrate afterwards with **unlabeled** [calibration data]?
> >
> > If our understanding is correct, won't this setup be unfair in the sense that model A has more supervised information than model B?
> > In this case, it appears that the "information budget" of these models differs. And it is even more severe, as here one model had access to more labeled data (which is crucial, as labeled data is typically hard to collect, in contrast to plain covariates).
> >
> > In practice, unlabeled data can be generated simply by augmenting the input covariates, whereas labeled data requires expert annotation.
> >
> > Once again, thank you for your reply. We look forward to the Reviewer's prompt response.

---

> > > ### Author Response · Authors · 2025-11-26
> > >
> > > Dear Reviewer qyiv,
> > >
> > > Following your suggestion, we conducted an experiment in which we aggregated different uncertainty measures in a straightforward additive way. We present the results for this experiment in the Table below.
> > > We used the same individual components as you suggested, as well as those used in the main text of the paper (sum of total, excess, Bayes risk, and Mahalanobis score).
> > >
> > > From the Table, we see that for CIFAR-10, the results are indeed close. However, this simple additive heuristic deteriorates on more complex datasets.
> > >
> > > The good behavior on CIFAR-10 for the simple baseline can be explained by the fact that each measure in the composition is already effective (see Tables 1 and 2 in the main text, which show performance for each 1D component). However, for CIFAR-100 and TinyImageNet, where the performance of each component begins to diverge, this simple aggregation strategy is not effective.
> > >
> > > Our VecUQ-OT aggregation consistently outperforms this simple additive baseline, and the difference becomes even more pronounced when metrics in the composition begin to disagree in their ordering.
> > >
> > >
> > > We hope that this experiment, performed as requested by the reviewer, resolves the concern (2).
> > >
> > > Regarding concern (3), this experiment also demonstrates that OT-based aggregation is beneficial, treating different components within the vector in a more nuanced manner than a simple, straightforward summation. We agree with the reviewer that experiments in the paper provide the empirical evidence that using the optimal transport approach is beneficial. We hope this also resolves the concern.
> > >
> > > We are happy to answer any questions the reviewer might have in this regard.
> > >
> > > We would also be grateful if the reviewer could respond to our earlier question about the proposed experimental setup. Having your feedback promptly will help us plan the experiments, obtain the results, and still have enough time to discuss them within the review process.
> > >
> > > Best regards,
> > >
> > > Authors
> > >
> > >
> > >
> > >
> > > | ind_dataset   | eval                     | naive_additive      | VecUQ-OT |
> > > |--------------|--------------------------|---------------------|----------------------------------------|
> > > | cifar10      | cifar10 [miscls]         | 0.931 ± 0.002       | 0.944 ± 0.002                          |
> > > | cifar10      | cifar10 [selective]      | 0.996 ± 0.000       | 0.997 ± 0.000                          |
> > > | cifar10      | cifar100 [ood]           | 0.914 ± 0.001       | 0.918 ± 0.001                          |
> > > | cifar10      | svhn [ood]               | 0.939 ± 0.004       | 0.957 ± 0.005                          |
> > > | cifar10      | tiny_imagenet [ood]      | 0.911 ± 0.001       | 0.912 ± 0.001                          |
> > > | cifar100     | cifar10 [ood]            | 0.554 ± 0.004       | 0.765 ± 0.001                          |
> > > | cifar100     | cifar100 [miscls]        | 0.598 ± 0.006       | 0.849 ± 0.003                          |
> > > | cifar100     | cifar100 [selective]     | 0.821 ± 0.001       | 0.916 ± 0.001                          |
> > > | cifar100     | svhn [ood]               | 0.703 ± 0.032       | 0.870 ± 0.006                          |
> > > | cifar100     | tiny_imagenet [ood]      | 0.690 ± 0.005       | 1.000 ± 0.000                          |
> > > | tiny_imagenet| tiny_imagenet [miscls]   | 0.431 ± 0.004       | 0.847 ± 0.002                          |
> > > | tiny_imagenet| tiny_imagenet [selective]| 0.672 ± 0.005       | 0.886 ± 0.001                          |
> > > | tiny_imagenet| imagenet_a [ood]         | 0.457 ± 0.009       | 0.847 ± 0.002                          |
> > > | tiny_imagenet| imagenet_r [ood]         | 0.419 ± 0.008       | 0.835 ± 0.001                          |
> > > | tiny_imagenet| imagenet_o [ood]         | 0.522 ± 0.005       | 0.760 ± 0.002                          |

---

> ### Author Response · Authors · 2025-11-26
>
> Dear Reviewer qyiv,
>
> We would like to kindly follow up on the last sentence of your previous response:
>
> > I would like to emphasize that the direction of this work is very interesting, but, in its current state, the work is not ready for acceptance.
>
> We are very grateful that you find the direction of the work interesting. This impression is also shared by other reviewers, who highlight the novelty [7cU2, LbkW] and creativity [D3gc] of the proposed approach.
>
> At the same time, we do not fully understand what, in your view, is still missing or insufficient in the *current state* of the paper.
>
> During the rebuttal, you did not point out specific problems with the writing or mathematics. Instead, you asked excellent and insightful questions about intuition and motivation, which we have tried to address carefully in our response.
>
> On the technical side, we:
> 1. Conducted two additional experiments with additive aggregation, directly addressing concern (2).
> 2. Clarified how one should choose the uncertainty measures in the vector and provided supporting empirical evidence, which (as we understood from your reply) addresses concern (3).
>
> Regarding concern (1) and the new experimental setup you proposed, our understanding is that it assumes one model has access to more labeled data.
> We worry that this may inherently favor certain methods in a way that makes the comparison less balanced. Of course, we may still be misunderstanding your intention here, and we would be very grateful for further clarification if that is the case.
>
> From our perspective, we have now responded to all the concrete concerns you explicitly raised. This is why we are somewhat unsure how to reconcile this with the overall score of “2: reject, not good enough.”
>
> If there are any additional issues that we have overlooked or if some of our clarifications were not convincing, we would be very grateful if you could point them out explicitly.
>
>
> Best regards,
>
> Authors

---

> ### Author Response · Authors · 2025-11-27
>
> Dear Reviewer qyiv,
>
>
> Following the question from Reviewer LbkW regarding the sensitivity of our OT algorithm to the size of calibration data, we conducted an additional experiment. In this experiment, we varied the size of the calibration data (100% corresponds to the results presented in the main text of the paper, with 1,800 objects for all datasets). See the Table below for the results.
>
> What we observe is that even for way smaller calibration dataset sizes, the method remains robust.
>
> We believe this also addresses your concern (1) on unfair comparison. Specifically, we significantly (10 or even 100 times) reduced the "data budget" the OT-algorithm sees (without labels). Yet, the results are still robust and close to the original (100%) ones.
>
> We are still willing to experiment with the scenario that the Reviewer considers fair. However, if our understanding is correct (see our response 2 days ago), the scenario the Reviewer has proposed allows one model to see more labeled data than another. In our opinion, this scenario cannot be fair.
>
> We thank the Reviewer and are looking for their response.
>
>
> Best regards,
>
> Authors
>
>
> -----
> -----
> -----
> -----
>
>
> | ind_dataset| eval| 100%| 70% | 50%| 20%| 10%| 1%| 0.5%|
> |-----|------|-----|----------------|----------------|----------------|----------------|----------------|----------------|
> | cifar10| cifar10 [miscls]| 0.944 ± 0.002  | 0.944 ± 0.002  | 0.944 ± 0.002  | 0.944 ± 0.002  | 0.944 ± 0.002  | 0.944 ± 0.002  | 0.941 ± 0.002  |
> | cifar10| cifar10 [selective]| 0.997 ± 0.000  | 0.997 ± 0.000  | 0.997 ± 0.000  | 0.997 ± 0.000  | 0.997 ± 0.000  | 0.997 ± 0.000  | 0.997 ± 0.000  |
> | cifar10| cifar100 [ood]| 0.918 ± 0.001  | 0.918 ± 0.001  | 0.918 ± 0.001  | 0.918 ± 0.001  | 0.918 ± 0.001  | 0.917 ± 0.002  | 0.916 ± 0.003  |
> | cifar10| svhn [ood]| 0.957 ± 0.005  | 0.957 ± 0.005  | 0.957 ± 0.005  | 0.958 ± 0.005  | 0.958 ± 0.005  | 0.956 ± 0.006  | 0.952 ± 0.006  |
> | cifar10| tiny_imagenet [ood]| 0.912 ± 0.001  | 0.912 ± 0.000  | 0.912 ± 0.000  | 0.913 ± 0.001  | 0.912 ± 0.001  | 0.911 ± 0.002  | 0.910 ± 0.004  |
> | cifar100| cifar10 [ood]| 0.765 ± 0.001  | 0.764 ± 0.001  | 0.763 ± 0.001  | 0.764 ± 0.002  | 0.760 ± 0.003  | 0.762 ± 0.003  | 0.634 ± 0.021  |
> | cifar100| cifar100 [miscls]| 0.849 ± 0.003  | 0.849 ± 0.002  | 0.848 ± 0.003  | 0.848 ± 0.004  | 0.846 ± 0.004  | 0.848 ± 0.002  | 0.703 ± 0.031  |
> | cifar100| cifar100 [selective]| 0.916 ± 0.001  | 0.915 ± 0.001  | 0.915 ± 0.001  | 0.915 ± 0.001  | 0.914 ± 0.001  | 0.916 ± 0.001  | 0.867 ± 0.010  |
> | cifar100| svhn [ood]| 0.870 ± 0.006  | 0.870 ± 0.006  | 0.868 ± 0.007  | 0.870 ± 0.007  | 0.864 ± 0.006  | 0.854 ± 0.006  | 0.738 ± 0.018  |
> | cifar100| tiny_imagenet [ood]| 1.000 ± 0.000  | 1.000 ± 0.000  | 1.000 ± 0.000  | 1.000 ± 0.000  | 1.000 ± 0.000  | 1.000 ± 0.000  | 0.995 ± 0.006  |
> | tiny_imagenet  | tiny_imagenet [miscls]| 0.847 ± 0.002  | 0.843 ± 0.002  | 0.841 ± 0.003  | 0.843 ± 0.003  | 0.839 ± 0.003  | 0.842 ± 0.004  | 0.720 ± 0.018  |
> | tiny_imagenet  | tiny_imagenet [selective] | 0.886 ± 0.001  | 0.884 ± 0.001  | 0.883 ± 0.001  | 0.884 ± 0.001  | 0.882 ± 0.002  | 0.885 ± 0.001  | 0.835 ± 0.007  |
> | tiny_imagenet  | imagenet_a [ood]| 0.847 ± 0.002  | 0.846 ± 0.002  | 0.845 ± 0.002  | 0.846 ± 0.002  | 0.843 ± 0.002  | 0.840 ± 0.005  | 0.744 ± 0.014  |
> | tiny_imagenet  | imagenet_r [ood]| 0.835 ± 0.001  | 0.832 ± 0.001  | 0.831 ± 0.001  | 0.832 ± 0.001  | 0.830 ± 0.001  | 0.828 ± 0.006  | 0.727 ± 0.016  |
> | tiny_imagenet  | imagenet_o [ood]| 0.760 ± 0.002  | 0.762 ± 0.001  | 0.763 ± 0.002  | 0.762 ± 0.002  | 0.763 ± 0.002  | 0.764 ± 0.003  | 0.712 ± 0.010  |

---

> ### Author Response · Authors · 2025-11-28
>
> Dear Reviewer qyiv,
>
> A few days ago, we submitted a detailed response to your comments, but we have not yet seen any updates from you. We would like to kindly ask you to take a look at our replies.
>
> In our responses, we addressed all of the concerns you raised. If anything remains unclear or unanswered, we would be very grateful if you could let us know so that we can clarify it.
>
> Thank you!
>
> Best regards,
>
> Authors

---

### Note · Authors · 2026-01-13

I have read and agree with the venue's withdrawal policy on behalf of myself and my co-authors.